# Exploring and Unleashing the Power of Message Passing on Heterophilous Graphs

## Abstract

Graph Neural Networks (GNNs) have demonstrated strong performance in graph mining tasks due to their message-passing mechanism, which is aligned with the homophily assumption that adjacent nodes exhibit similar behaviors. However, in many real-world graphs, connected nodes may display contrasting behaviors, termed as *heterophilous* patterns, which has attracted increased interest in heterophilous GNNs (HTGNNs). Although the message-passing mechanism seems unsuitable for heterophilous graphs due to the propagation of class-irrelevant information, it is still widely used in many existing HTGNNs and consistently achieves notable success. This raises the question: *why does message passing remain effective on heterophilous graphs?* To answer this question, in this paper, we revisit the message-passing mechanisms in heterophilous graph neural networks and reformulate them into a unified heterophilious message-passing (HTMP) mechanism. Based on HTMP and empirical analysis, we reveal that the success of message passing in existing HTGNNs is attributed to implicitly enhancing the compatibility matrix among classes. Moreover, we argue that the full potential of the compatibility matrix is not completely achieved due to the existence of incomplete and noisy semantic neighborhoods in real-world heterophilous graphs. To bridge this gap, we introduce a new approach named CMGNN, which operates within the HTMP mechanism to explicitly leverage and improve the compatibility matrix. A thorough evaluation involving 10 benchmark datasets and comparative analysis against 17 well-established baselines highlights the superior performance of the HTMP mechanism and CMGNN method.

## 1 Introduction

Graph Neural Networks (GNNs) have shown remarkable performance in graph mining tasks, such as social network analysis (Kipf & Welling, 2017; Zhang et al., 2022) and recommender systems (Wang et al., 2019; He et al., 2020). The design principle of GNNs is typically based on the homophily assumption (McPherson et al., 2001), which assumes that nodes are inclined to exhibit behaviors similar to their neighboring nodes (Ma et al., 2022). However, this assumption does not always hold in real-world graphs, where the connected nodes demonstrate a contrasting tendency known as the *heterophily* (Zhu et al., 2021a). In response to the challenges of heterophily in graphs, *heterophilous GNNs (HTGNNs)* have attracted considerable research interest (Ma et al., 2022; Zheng et al., 2022; Zhu et al., 2023), with numerous innovative approaches being introduced recently, such as Abu-El-Haija et al. (2019); Bo et al. (2021); Luan et al. (2022); Song et al. (2023). However, the majority of these methods continue to employ a message-passing mechanism, which was not originally designed for heterophilous graphs, as they tend to incorporate excessive information from disparate classes. This naturally raises a question: *Why does message passing remain effective on heterophilous graphs?*

Recently, a few efforts (Ma et al., 2022; Zhu et al., 2023) have begun to investigate this question and reveal that vanilla message passing can work on heterophilous graphs under certain conditions. However, the absence of a unified and comprehensive understanding of message passing within existing HTGNNs has hindered the creation of innovative approaches. In this paper, we first revisit the message-passing mechanisms in existing HTGNNs and reformulate them into a unified heterophilous message-passing (HTMP) mechanism, which extends the definition of neighborhood in various ways and simultaneously utilizes the messages of multiple neighborhoods. Specifically, HTMP consists of

three major steps namely aggregating messages with explicit guidance, combining messages from multiple neighborhoods, and fusing intermediate representations.

Equipped with HTMP, we further conduct empirical analysis on real-world graphs. The results reveal that the success of message passing in existing HTGNNs is attributed to *implicitly enhancing the compatibility matrix*, which exhibits the probabilities of observing edges among nodes from different classes. In particular, by increasing the distinctiveness between the rows of the compatibility matrix via different strategies, the node representations of different classes become more discriminative.

Drawing from previous observations, we contend that nodes within real-world graphs might exhibit a semantic neighborhood that only reveals a fraction of the compatibility matrix, accompanied by noise. This could limit the effectiveness of enhancing the compatibility matrix and result in suboptimal representations. To fill this gap, we further propose a novel Compatibility Matrix-aware Graph Neural Network (CMGNN) under HTMP mechanism, which utilizes the compatibility matrix to construct desired neighborhood messages as supplementary for nodes and explicitly enhances the compatibility matrix by a targeted constraint. We build a benchmark to fairly evaluate CMGNN and existing methods, which encompasses 17 diverse baseline methods and 10 datasets that exhibit varying levels of heterophily. Extensive experimental results demonstrate the superiority of CMGNN and HTMP mechanism. The contributions of this paper are summarized as follows:

- We revisit the message-passing mechanisms in existing HTGNNs and reformulate them into a unified heterophilous message-passing mechanism (HTMP), which not only provides a macroscopic view of message passing in HTGNNs but also enables people to develop new methods flexibly.

- We reveal that the effectiveness of message passing on heterophilous graphs is attributed to implicitly enhancing the compatibility matrix among classes, which gives us a new perspective to understand the message passing in HTGNNs.

- Based on HTMP mechanism and empirical analysis, we propose CMGNN to unlock the potential of the compatibility matrix in HTGNNs. We further build a unified benchmark that avoids the issues of current datasets for fair evaluation[1]. Experiments show the superiority of CMGNN.

## 2 PRELIMINARIES

Given a graph $\mathcal{G} = (\mathcal{V}, \mathcal{E}, \mathbf{X}, \mathbf{A}, \mathbf{Y})$, $\mathcal{V}$ is the node set and $\mathcal{E}$ is the edge set. Nodes are characterized by the feature matrix $\mathbf{X} \in \mathbb{R}^{N \times d_f}$, where $N = |\mathcal{V}|$ denotes the number of nodes, $d_f$ is the features dimension. $\mathbf{Y} \in \mathbb{R}^{N \times 1}$ is the node labels with the one-hot version $\mathbf{C} \in \mathbb{R}^{N \times K}$, where $K$ is the number of node classes. The neighborhood of node $v_i$ is denoted as $\mathcal{N}_i$. $\mathbf{A} \in \mathbb{R}^{N \times N}$ is the adjacency matrix , and $\mathbf{D} = \text{diag}(\mathbf{d}_1, ..., \mathbf{d}_n)$ represents the diagonal degree matrix, where $\mathbf{d}_i = \sum_j \mathbf{A}_{ij}$. $\tilde{\mathbf{A}} = \mathbf{A} + \mathbf{I}$ represents the adjacency matrix with self-loops. Let $\mathbf{Z} \in \mathbb{R}^{N \times d_r}$ be the node representations with dimension $d_r$ learned by the models. We use $\mathbf{1}$ to represent a matrix with all elements equal to 1, and $\mathbf{0}$ for a matrix with all elements equal to 0.

**Homophily and Heterophily**. High homophily is observed in graphs where a substantial portion of connected nodes shares identical labels, while high heterophily corresponds to the opposite situation. For measuring the homophily level, two widely used metrics are edge homophily $h^e$ (Zhu et al., 2020) and node homophily $h^n$ (Pei et al., 2020), defined as $h^e = \frac{|\{e_{u,v}|e_{u,v} \in \mathcal{E}, \mathbf{Y}_u = \mathbf{Y}_v\}|}{|\mathcal{E}|}$ and $h^n = \frac{1}{|\mathcal{V}|} \sum_{v \in \mathcal{V}} \frac{|\{u|u \in \mathcal{N}_v, \mathbf{Y}_u = \mathbf{Y}_v\}|}{\mathbf{d}_v}$. Both metrics have a range of $[0, 1]$, where higher values indicate stronger homophily and lower values indicate stronger heterophily.

**Vanilla Message Passing (VMP)**. The vanilla message-passing mechanism plays a pivotal role in transforming and updating node representations based on the neighborhood (Gilmer et al., 2017). Typically, the mechanism operates iteratively and comprises two stages:

$$\widetilde{\mathbf{Z}}^l = \text{AGGREGATE}(\mathbf{A}, \mathbf{Z}^{l-1}), \quad \mathbf{Z}^l = \text{COMBINE}\left(\mathbf{Z}^{l-1}, \widetilde{\mathbf{Z}}^l\right), \tag{1}$$

where the AGGREGATE function first aggregates the input messages $\mathbf{Z}^{l-1}$ from neighborhood $\mathbf{A}$ into the aggregated one $\widetilde{\mathbf{Z}}^l$, and subsequently, the COMBINE function combines the messages of node ego and neighborhood aggregation, resulting in updated representations $\mathbf{Z}^l$.

---

[1]Codebase is available in the supplementary material.

Table 1: Revisiting the message passing in representative heterophilous GNNs under the perspective of HTMP mechanism.

| Method | Neighborhood Indicators | | Aggregation Guidance | | COMBINE | FUSE |
|---|---|---|---|---|---|---|
| | Type | $\mathcal{A}$ | Type | $\mathcal{B}$ | | |
| GCN (Kipf & Welling, 2017) | Raw | $[\tilde{\mathbf{A}}]$ | DegAvg | $[\tilde{\mathbf{B}}^d]$ | / | $\mathbf{Z}=\mathbf{Z}^L$ |
| APPNP (Gasteiger et al., 2019) | | $[\mathbf{I},\tilde{\mathbf{A}}]$ | | $[\mathbf{I},\tilde{\mathbf{B}}^d]$ | WeightedAdd | $\mathbf{Z}=\mathbf{Z}^L$ |
| GCNII (Chen et al., 2020) | | $[\mathbf{I},\tilde{\mathbf{A}}]$ | | $[\mathbf{I},\tilde{\mathbf{B}}^d]$ | WeightedAdd | $\mathbf{Z}=\mathbf{Z}^L$ |
| GAT (Veličković et al., 2018) | | $[\tilde{\mathbf{A}}]$ | AdaWeight | $[\mathbf{B}^{aw}]$ | / | $\mathbf{Z}=\mathbf{Z}^L$ |
| GPR-GNN (Chien et al., 2021) | | $[\tilde{\mathbf{A}}]$ | | $[\tilde{\mathbf{B}}^d]$ | / | AdaAdd |
| OrderedGNN (Song et al., 2023) | | $[\mathbf{I},\mathbf{A}]$ | DegAvg | $[\mathbf{I},\mathbf{B}^d]$ | AdaCat | $\mathbf{Z}=\mathbf{Z}^L$ |
| ACM-GCN (Luan et al., 2022) | | $[\mathbf{I},\mathbf{A},\tilde{\mathbf{A}}]$ | | $[\mathbf{I},\mathbf{B}^d,\mathbf{I}-\mathbf{B}^d]$ | AdaAdd | $\mathbf{Z}=\mathbf{Z}^L$ |
| FAGCN (Bo et al., 2021) | | $[\mathbf{I},\mathbf{A}]$ | AdaWeight | $[\mathbf{I},\mathbf{B}^{naw}]$ | WeightedAdd | $\mathbf{Z}=\mathbf{Z}^L\mathbf{W}$ |
| GBK-GNN (Du et al., 2022) | | $[\mathbf{I},\mathbf{A},\mathbf{A}]$ | | $[\mathbf{I},\mathbf{B}^{aw},\mathbf{1}-\mathbf{B}^{aw}]$ | Add | $\mathbf{Z}=\mathbf{Z}^L$ |
| SimP-GCN (Jin et al., 2021b) | ReDef | $[\mathbf{I},\tilde{\mathbf{A}},\mathbf{A}_f]$ | | $[\mathbf{I},\tilde{\mathbf{B}}^d,\mathbf{B}_f^d]$ | AdaAdd | $\mathbf{Z}=\mathbf{Z}^L$ |
| H2GCN (Zhu et al., 2020) | | $[\mathbf{A},\mathbf{A}_{h2}]$ | DegAvg | $[\mathbf{B}^d,\mathbf{B}_{h2}^d]$ | Cat | Cat |
| Geom-GCN (Pei et al., 2020) | | $[\mathbf{A}_{c1},...,\mathbf{A}_{cr},...,\mathbf{A}_{cR}]$ | | $[\mathbf{B}_{c1}^d,...,\mathbf{B}_{cr}^d,...,\mathbf{B}_{cR}^d]$ | Cat | $\mathbf{Z}=\mathbf{Z}^L$ |
| MixHop (Abu-El-Haija et al., 2019) | | $[\mathbf{I},\mathbf{A},\mathbf{A}_{h2},...,\mathbf{A}_{hk}]$ | | $[\mathbf{I},\mathbf{B}^d,\mathbf{B}_{h2}^d,...,\mathbf{B}_{hk}^d]$ | Cat | $\mathbf{Z}=\mathbf{Z}^L$ |
| UGCN (Jin et al., 2021a) | | $[\tilde{\mathbf{A}},\tilde{\mathbf{A}}_{h2},\mathbf{A}_f]$ | AdaWeight | $[\tilde{\mathbf{B}}^{aw},\tilde{\mathbf{B}}_{h2}^{aw},\mathbf{B}_f^{aw}]$ | AdaAdd | $\mathbf{Z}=\mathbf{Z}^L$ |
| WRGNN (Suresh et al., 2021) | | $[\mathbf{A}_{c1},...,\mathbf{A}_{cr},...,\mathbf{A}_{cR}]$ | | $[\mathbf{B}_{c1}^{aw},...,\mathbf{B}_{cr}^{aw},...,\mathbf{B}_{cR}^{aw}]$ | Add | $\mathbf{Z}=\mathbf{Z}^L$ |
| HOG-GCN (Wang et al., 2022) | | $[\mathbf{I},\mathbf{A}_{hk}]$ | | $[\mathbf{I},\mathbf{B}^{re}]$ | WeightedAdd | $\mathbf{Z}=\mathbf{Z}^L$ |
| GloGNN (Li et al., 2022) | | $[\mathbf{I},\mathbf{1}]$ | RelaEst | $[\mathbf{I},\mathbf{B}^{re}]$ | WeightedAdd | $\mathbf{Z}=\mathbf{Z}^L$ |
| GGCN (Yan et al., 2022) | Dis | $[\mathbf{I},\mathbf{A}_p,\mathbf{A}_n]$ | | $[\mathbf{I},\mathbf{B}_p^{re},\mathbf{B}_n^{re}]$ | AdaAdd | $\mathbf{Z}=\mathbf{Z}^L$ |

* The correspondence between the full form and the abbreviation: Raw Neighborhood (Raw), Neighborhood Redefine (ReDef), Neighborhood Discrimination (Dis), Degree-based Averaging (DegAvg), Adaptive Weights (AdaWeight), Relation Estimation (RelaEst), Addition (Add), Weighted Addition (WeightAdd), Adaptive Weighted Addition (AdaAdd), Concatenation (Cat), Adaptive Dimension Concatenation (AdaCat).
* More details about the notations are available in Appendix B.1.

# 3 REVISITING MESSAGE PASSING IN HETEROPHILOUS GNNs.

To gain a thorough and unified insight into the effectiveness of message passing in HTGNNs, we revisit message passing in various notable HTGNNs (Bo et al., 2021; Zhu et al., 2020; Jin et al., 2021a;b; Pei et al., 2020; Abu-El-Haija et al., 2019; Wang et al., 2022; Luan et al., 2022; Li et al., 2022; Chien et al., 2021; Song et al., 2023; Suresh et al., 2021; Yan et al., 2022; Du et al., 2022) and propose a unified heterophilous message passing (HTMP) mechanism, structured as follows:

$$\widetilde{\mathbf{Z}}_r^l = \text{AGGREGATE}(\mathbf{A}_r,\mathbf{B}_r,\mathbf{Z}^{l-1}),\ \mathbf{Z}^l = \text{COMBINE}(\{\widetilde{\mathbf{Z}}_r^l\}_{r=1}^R),\ \mathbf{Z} = \text{FUSE}(\{\mathbf{Z}^l\}_{l=0}^L). \quad (2)$$

Generally, HTMP extends the definition of neighborhood in various ways and simultaneously utilizes the messages of multiple neighborhoods, which is the key to better adapting to heterophily. We use $R$ to denote the number of neighborhoods used by the model. In each message passing layer $l$, HTMP separately aggregates messages within $R$ neighborhoods and combines them. The methodological analysis of some representative HTGNNs and more details can be seen in Appendix B. Compared to the VMP mechanism, HTMP mechanism has progressed in the following functions:

(i) To characterize different neigborhoods, the **AGGREGATE** function in HTMP includes the **neighborhood indicator** $\mathbf{A}_r$ to indicate the neighbors within a specific neighborhood $r$. The adjacency matrix $\mathbf{A}$ in VMP is a special neighborhood indicator that marks the neighbors in the raw neighborhood. To further characterize the aggregation of different neighborhoods, HTMP introduces the **aggregation guidance** $\mathbf{B}_r$ for each neighborhood $r$. In VMP, the aggregation guidance is an implicit parameter of the AGGREGATE function since it only works for the raw neighborhood. A commonly used form of the AGGREGATE function is $\text{AGGREGATE}(\mathbf{A}_r,\mathbf{B}_r,\mathbf{Z}^{l-1}) = (\mathbf{A}_r \odot \mathbf{B}_r)\mathbf{Z}^{l-1}\mathbf{W}_r^l$, where $\odot$ is the Hadamard product and $\mathbf{W}_r^l$ is a weight matrix for message transformation. We take this as the general form of the AGGREGATE function and only analyze the neighborhood indicators and the aggregation guidance in the following.

The *neighborhood indicator* $\mathbf{A}_r \in \{0,1\}^{N\times N}$ indicates neighbors associated with central nodes within neighborhood $r$. To describe the multiple neighborhoods in HTGNNs, neighborhood indicators can be formed as a list $\mathcal{A} = [\mathbf{A}_1,...,\mathbf{A}_r,...,\mathbf{A}_R]$. For the sake of simplicity, we consider the identity matrix $\mathbf{I} \in \mathbb{R}^{N\times N}$ as a special neighborhood indicator for acquiring the ego messages of central nodes. The *aggregation guidance* $\mathbf{B}_r \in \mathbb{R}^{N\times N}$ can be viewed as pairwise aggregation weights in most cases, which has the multiple form $\mathcal{B} = [\mathbf{B}_1,...,\mathbf{B}_r,...,\mathbf{B}_R]$. Table 1 illustrates the connection between message passing in various HTGNNs and HTMP mechanism.

(ii) Considering the existence of multiple neighborhoods, the **COMBINE** function in HTMP need to integrate multiple messages instead of only the ego node and the raw neighborhood. Thus, the input of the COMBINE function is a set of messages $\widetilde{\mathbf{Z}}_r^l$ aggregated from the corresponding neighborhoods. In HTGNNs, addition and concatenation are two common approaches, each of which has variants. An effective COMBINE function is capable of simultaneously processing messages from various neighborhoods while preserving their distinct features, thereby reducing the effects of heterophily.

(iii) In VMP, the final output representations are usually one of the final layers: $\mathbf{Z} = \mathbf{Z}^L$. Some HTGNNs utilize the combination of intermediate representations to leverage messages from different localities, adapting to the heterophilous structural properties in different graphs. Thus, we introduce an additional **FUSE** function in HTMP which integrates multiple representations $\mathbf{Z}^l$ of different layers $l$ into the final $\mathbf{Z}$. Similarly, the FUSE function is based on addition and concatenation.

## 4 WHY DOES MESSAGE PASSING REMAIN EFFECTIVE IN HETEROPHILOUS GRAPHS?

Based on HTMP mechanism, we further dive into the motivation behind the message passing of existing HTGNNs. Our discussion begins by examining the difference between homophilous and heterophilous graphs. Initially, we consider homophily ratios $h^e$ and $h^n$, as outlined in Section 2. However, a single number can not indicate enough conditions for a graph. Ma et al. (2022) propose the existence of a special case of heterophily, named *"good" heterophily*, where the homophily ratios stay low but the VMP mechanism can achieve strong performance. Thus, to better study the heterophily property, we introduce the *Compatibility Matrix* (Zhu et al., 2021a) to describe graphs:

**Definition 1.** *Compatibility Matrix (CM): The potential connection preference among classes within a graph. It is formatted as a matrix $\mathbf{M} \in \mathbb{R}^{K \times K}$, where the $i$-th row $\mathbf{M}_i$ denotes the connection probabilities between class $i$ and all classes. It can be estimated empirically as follows:*

$$\mathbf{M} = Norm(\mathbf{C}^T \mathbf{C}^{nb}), \quad \mathbf{C}^{nb} = \hat{\mathbf{A}}\mathbf{C}, \tag{3}$$

*where $Norm(\cdot)$ denotes the L1 normalization for matrix row vectors and $T$ is the matrix transpose operation. $\mathbf{C}^{nb} \in \mathbf{R}^{N \times K}$ is the **semantic neighborhoods** of nodes, which indicates the proportion of neighbors from each class in the neighborhoods.*

We first visualize the CM of a homophilous graph Photo (Shchur et al., 2018) in Figure 1. It displays an identity-like matrix, where the diagonal elements can be viewed as the homophily level of each class. With this type of CM, the VMP mechanism learns representations comprised mostly of messages from same the class, while messages of other classes are diluted.

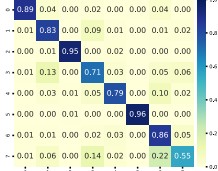

*Then how does HTMP mechanism work on heterophilous graphs with potentially chaotic CMs such as Amazon-Ratings (Platonov et al., 2023) in Figure 2(a)?* The "good" heterophily inspires us, which we believe corresponds to a CM with enough discriminability among classes. We conduct experiments on synthetic graphs to confirm this idea, with details available in Appendix C. Also, we find "good" heterophily exists in real-world graphs though it is not as significant as imagined. As a result, we have the following observation:

Figure 1: Observed CM of Photo.

**Observation 1.** *(Connection between CM and VMP). When enough (depends on data) discriminability exists among classes in CM, vanilla message passing can work well in heterophilous graphs.*

This observation is similar to some prior works (Ma et al., 2022; Zhu et al., 2023) which emphasize data while our focus is more on the message passing. Further, we have the following theorem with detailed proof in Appendix D:

**Theorem 1.** *The discriminability among the representations learned by the message-passing mechanism is **positively correlated** with the discriminability among classes in the compatibility matrix.*

Based on Theorem 1, we have a conjecture about the reason for HTMP's effectiveness: *The HTMP mechanism tries to enhance the discriminability of CM, which contributes to better representations*. Some special designs in HTMP intuitively meet this. For example, *feature-similarity-based neighborhood indicators* and *neighborhood discrimination* are designed to construct neighborhoods

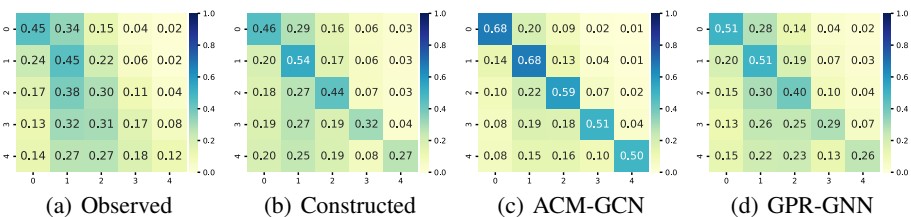

Figure 2: Visualizations of the compatibility matrices of Amazon-Ratings.

with high homophily, that is, an identity-like CM with high discriminability. We plot the CM of constructed feature-similarity-based neighborhood on Amazon-Ratings in Figure 2(b) to confirm it. Moreover, we investigate two representative methods ACM-GCN (Luan et al., 2022) and GPR-GNN (Chien et al., 2021), showing that they also meet this conjecture with the posterior evidence in Figure 2(c) and 2(d), which demonstrates that they have enhanced the discriminability of CM. More details about the posterior proof are available in Appendix E. ACM-GCN combines the messages from different filters with adaptive weights, which actually modifies the edge and node weights to build a new CM. GPR-GNN has a FUSE function that integrates the CMs of multiple-order neighborhoods with adaptive weights to form a more discriminative CM. These evidences lead to the answer to the aforementioned question:

**Observation 2.** *(Connection between CM and HTMP). The unified goal of various message passing in existing HTGNNs is to utilize and enhance the discriminability of CM on heterophilous graphs. In other words, the success of message passing in existing HTGNNs benefits from utilizing and enhancing the discriminability of CM.*

Furthermore, we notice that the power of CM is not fully released due to the incomplete and noisy semantic neighborhoods in real-world heterophilous graphs. We use the perspective of distribution to describe the issue more intuitively: The semantic neighborhoods of nodes from the same class collectively form a distribution, whose mean value indicates the connection preference of that class, i.e. $\mathbf{M}_i$ for class $i$. Influenced by factors such as degree and randomness, the semantic neighborhood of nodes in real-world graphs may display only a fraction of CM accompanied by noise. It can lead to the overlap between different distributions as shown in Figure 4, where the existence of overlapping parts means nodes from different classes may have the same semantic neighborhood. This brings a great challenge since the overlapping semantic neighborhood may become redundant information during message passing.

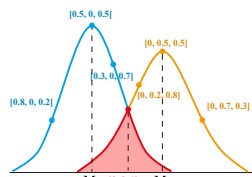

Figure 3: Overlap of semantic neighborhood distribution.

## 5 METHODOLOGY

To fill this gap, we further propose a method named Compatibility Matrix-Aware GNN (CMGNN) as shown in Figure 4, which leverages the CM to construct desired neighborhood messages as supplementary, providing valuable neighborhood information for nodes to mitigate the impact of incomplete and noisy semantic neighborhoods. Thus, we first construct supplementary neighborhoods for all nodes to guarantee the accessibility of messages from all classes. CMGNN follows the HTMP mechanism and constructs a supplementary neighborhood indicator along with the corresponding aggregation guidance to introduce supplementary messages. Further, CMGNN introduces a simple constraint to explicitly enhance the discriminability of CM.

**Supplementary Neighborhood Construction** CMGNN introduces supplementary neighborhoods to provide nodes with messages from each class. The supplementary neighborhood indicator $\mathbf{A}^{sup}$ assigns $K$ additional virtual neighbors for each node: $\mathbf{A}^{sup} = \mathbf{1} \in \mathbb{R}^{N \times K}$. Specifically, these additional neighbors are $K$ virtual nodes, constructed as the prototypes of classes based on the labels of the training set. Considering the sparsity of graphs, some nodes may have low degrees. Thus, the all-one neighborhood indicator $\mathbf{A}^{sup}$ guarantees the accessibility to the messages from each class for

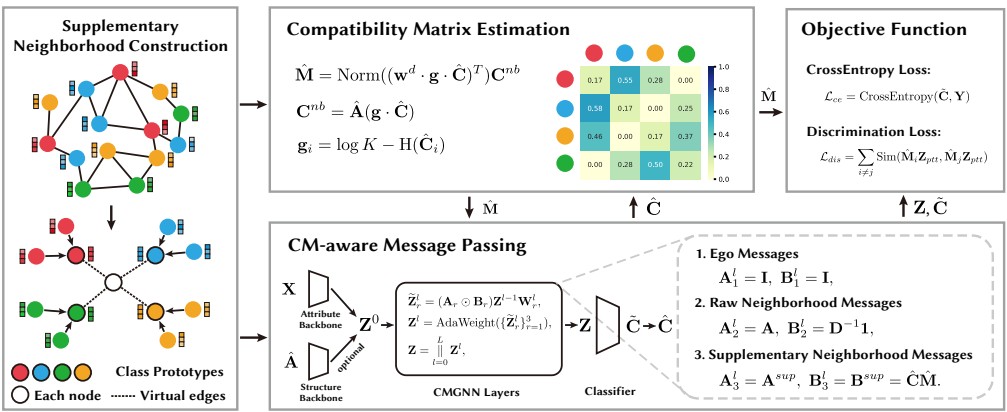

Figure 4: The overall framework of CMGNN. It contains three main parts: (1) Supplementary Neighborhood Construction, which constructs class prototypes as additional virtual neighbors for all nodes; (2) Compatibility Matrix Estimation and (3) CM-aware Message Passing. Parts (2) and (3) are iterative as higher-quality predictions produce more accurate CM and vice versa.

**all nodes.** The attributes $\mathbf{X}^{ptt} \in \mathbb{R}^{K \times d_f}$, neighborhoods $\mathbf{A}^{ptt} \in \mathbb{R}^{K \times N}$ and labels $\mathbf{Y}^{ptt} \in \mathbb{R}^{K \times K}$ of prototypes are defined as follows:

$$\mathbf{X}^{ptt} = \text{Norm}(\mathbf{C}_{train}{}^T \mathbf{X}_{train}), \ \mathbf{A}^{ptt} = \mathbf{0}, \ \mathbf{Y}^{ptt} = \mathbf{I}, \tag{4}$$

where $\mathbf{C}_{train}$ and $\mathbf{X}_{train}$ are the one-hot labels and attributes of nodes in the training set. Utilizing class prototypes as supplementary neighborhoods can provide each node with representative messages of classes, which builds the basis for desired neighborhood messages.

**Compatibility Matrix Estimation.** The CM can be directly calculated via Eq 3 with full-available labels. However, the label information is not entirely available in semi-supervised settings. Thus, we try to estimate the CM with the help of semi-supervised and pseudo labels. Since the pseudo labels predicted by the model might be wrong, which can lead to low-quality estimation, we introduce the confidence $\mathbf{g} \in \mathbb{R}^{N \times 1}$ based on the information entropy to reduce the impact of wrong predictions, where a high entropy means low confidence:

$$\mathbf{g}_i = \log K - \text{H}(\hat{\mathbf{C}}_i) \in [0, \log K], \tag{5}$$

where $\text{H}(p) = -\sum_i p_i \log(p_i)$ denotes the entropy, $\hat{\mathbf{C}} \in \mathbb{R}^{N \times K}$ is the soft pseudo labels composed of training labels $\mathbf{C}_{train}$ and model predictions $\tilde{\mathbf{C}}$ which is introduced later:

$$\hat{\mathbf{C}}_i = \begin{cases} \mathbf{C}_{train,i}, & v_i \in \mathcal{V}_{train}, \\ \tilde{\mathbf{C}}_i, & \text{otherwise,} \end{cases} \tag{6}$$

where $\mathcal{V}_{train}$ denotes the training set. Then the semantic neighborhoods of the nodes are calculated considering the confidence: $\mathbf{C}^{nb} = \text{Norm}(\mathbf{A}(\mathbf{g} \cdot \hat{\mathbf{C}})) \in \mathbb{R}^{N \times K}$.

In addition, the degrees of nodes also influence the estimation. As mentioned in Section 4, the semantic neighborhood of low-degree nodes may display incomplete CM, leading to a significant gap between semantic neighborhoods and corresponding CM. Thus, nodes with low degrees deserve low weights during the estimation. We manually set up a weighting function range in $[0, 1]$:

$$\mathbf{w}_i^d = \begin{cases} \mathbf{d}_i/2K, & \mathbf{d}_i \leq K, \\ 0.25 + \mathbf{d}_i/4K, & K < \mathbf{d}_i \leq 3K, \\ 1, & otherwise. \end{cases} \tag{7}$$

For low-degree nodes, increases in degree should yield more significant benefits compared to high-degree nodes. Beyond a certain threshold, increases in degree yield tiny benefits. We have empirically chosen $K$ and $3K$ as fixed thresholds for the weighting function to simplify the design without multiple attempts. This approach is straightforward and can be substituted with other forms that meet the same criteria. Finally, we can estimate the compatibility matrix $\hat{\mathbf{M}} \in \mathbb{R}^{K \times K}$ as follows:

$$\hat{\mathbf{M}} = \text{Norm}((\mathbf{w}^d \cdot \mathbf{g} \cdot \hat{\mathbf{C}})^T)\mathbf{C}^{nb}. \tag{8}$$

Note that CM is repeatedly updated during the training. For the sake of efficiency, we do not estimate CM in each epoch. Instead, we save it as fixed parameters and only update it when the evaluation performance is improved.

**CM-aware Message Passing**   CMGNN aggregates messages from three neighborhoods for each node, including the ego, raw, and supplementary neighborhoods. The first two are the most commonly used and contain information about the central node itself and its neighbors respectively, while the latter is a new way to utilize CM. The ego neighborhood contains messages of only node ego regardless of any neighbors, which can be formatted as follows:

$$\mathbf{A}_1^l = \mathbf{I}, \quad \mathbf{B}_1^l = \mathbf{I}, \quad \mathbf{Z}_1^{l-1} = \mathbf{Z}^{l-1}, \quad \widetilde{\mathbf{Z}}_1^l = (\mathbf{A}_1^l \odot \mathbf{B}_1^l)\mathbf{Z}_1^{l-1}\mathbf{W}_1^l = \mathbf{Z}^{l-1}\mathbf{W}_1^l. \tag{9}$$

The raw neighborhood contains messages of raw neighbors without node ego, which meets the ego-neighbor separation design principle Zhu et al. (2020):

$$\mathbf{A}_2^l = \mathbf{A}, \quad \mathbf{B}_2^l = \mathbf{D}^{-1}\mathbf{1}, \quad \mathbf{Z}_2^{l-1} = \mathbf{Z}^{l-1}, \quad \widetilde{\mathbf{Z}}_2^l = (\mathbf{A}_2^l \odot \mathbf{B}_2^l)\mathbf{Z}_2^{l-1}\mathbf{W}_2^l = \mathbf{D}^{-1}\mathbf{A}\mathbf{Z}^{l-1}\mathbf{W}_2^l. \tag{10}$$

The supplementary neighborhood leverages CM to provide nodes with desired neighborhood messages, which implies the averaging message within a neighborhood when a node's semantic neighborhoods meet the CM of the corresponding class, converting the discriminability from CM into messages. It can be formatted as follows:

$$\mathbf{A}_3^l = \mathbf{A}^{sup}, \quad \mathbf{B}_3^l = \mathbf{B}^{sup} = \hat{\mathbf{C}}\hat{\mathbf{M}}, \quad \mathbf{Z}_3^{l-1} = \mathbf{Z}_{ptt}^{l-1},$$
$$\widetilde{\mathbf{Z}}_3^l = (\mathbf{A}_3^l \odot \mathbf{B}_3^l)\mathbf{Z}_3^{l-1}\mathbf{W}_3^l = (\mathbf{A}^{sup} \odot \hat{\mathbf{C}}\hat{\mathbf{M}})\mathbf{Z}_{ptt}^{l-1}\mathbf{W}_3^l, \tag{11}$$

where $\mathbf{Z}_{ptt}^{l-1}$ are the representations of virtual prototype nodes, obtained by the same message-passing mechanism as real nodes. The supplementary aggregation guidance $\mathbf{B}^{sup}$ indicates the desired semantic neighborhood of nodes, i.e. the desired proportion of neighbors from each class according to the probability that nodes belong to each class. Using soft logits instead of one-hot pseudo labels preserves the real characteristics of nodes and reduces the impact of wrong predictions.

Considering the various situations of different nodes, we use adaptive weighted addition to combine the messages from the above three neighborhoods. Meanwhile, the messages of multiple layers are concatenated to reserve the information with different locality in the graph. In the perspective of HTMP mechanism, the message passing of CMGNN cen be described as follows:

$$\widetilde{\mathbf{Z}}_r^l = \text{AGGREGATE}(\mathbf{A}_r, \mathbf{B}_r, \mathbf{Z}_r^{l-1}) = (\mathbf{A}_r \odot \mathbf{B}_r)\mathbf{Z}_r^{l-1}\mathbf{W}_r^l,$$
$$\mathbf{Z}^l = \text{COMBINE}(\{\widetilde{\mathbf{Z}}_r^l\}_{r=1}^3) = \text{AdaWeight}(\{\widetilde{\mathbf{Z}}_r^l\}_{r=1}^3), \tag{12}$$
$$\mathbf{Z} = \text{FUSE}(\{\mathbf{Z}^l\}_{l=0}^L) = \mathop{\|}_{l=0}^L \mathbf{Z}^l,$$

where AdaWeight is the adaptive weighted addition, $\|$ denotes the concatenation. Similar to existing methods (Luan et al., 2022; Li et al., 2022), we regard topology structure as additional available node features, which are the connection relationship among nodes, represented by the adjacency matrix $\mathbf{A}$. Each row $\mathbf{A}_i$ can be viewed as an additional $N$-dimensional feature of the corresponding node $i$. Thus, the input representation of the first layer can be obtained in two ways:

$$\mathbf{Z}^0 = [\mathbf{X}\mathbf{W}^X \| \hat{\mathbf{A}}\mathbf{W}^A]\mathbf{W}^0, \text{ or } \mathbf{Z}^0 = \mathbf{X}\mathbf{W}^0. \tag{13}$$

Specifically, (i) using additional features, where $\mathbf{W}^X \in \mathbb{R}^{d_f \times d_r}$, $\mathbf{W}^A \in \mathbb{R}^{N \times d_r}$ and $\mathbf{W}_0 \in \mathbb{R}^{2d_r \times d_r}$ are learnable matrices; (ii) using only attribute features, where $\mathbf{W}^0 \in \mathbb{R}^{d_f \times d_r}$. In practice, we use ReLU as the activation function between layers. From the perspective of HTMP mechanism, our special design is to introduce an additional neighborhood indicator $\mathbf{A}^{sup}$ by neighborhood redefining and aggregation guidance $\mathbf{B}^{sup}$, which can be seen as a form of relation estimation with good interpretability. Meanwhile, these designs require low time and space costs by the $N \times K$ form.

The prediction of the model is utilized during message passing. For initialization, nodes have the same probabilities belonging to each class. During the message passing, the predicted soft label $\tilde{\mathbf{C}}$ is replaced by the output of CMGNN, formatted as follow:

$$\tilde{\mathbf{C}} = \text{CLA}(\mathbf{Z}), \tag{14}$$

where CLA is a classifier implemented by an MLP and $\mathbf{Z}$ is the final node representation.

Table 2: Node classification accuracy comparison (%). The error bar (±) denotes the standard deviation of results over 10 trial runs. The best and second-best results in each column are highlighted in **bold** font and underlined. OOM denotes out-of-memory error during the model training.

| Dataset | Roman-Empire | Amazon-Ratings | Chameleon-F | Squirrel-F | Actor | Flickr | BlogCatalog | Wikics | Pubmed | Photo | Avg. Rank |
|---|---|---|---|---|---|---|---|---|---|---|---|
| **Homo.** | 0.05 | 0.38 | 0.25 | 0.22 | 0.22 | 0.24 | 0.4 | 0.65 | 0.8 | 0.83 | |
| **Nodes** | 22,662 | 24,492 | 890 | 2,223 | 7,600 | 7,575 | 5,196 | 11,701 | 19,717 | 7,650 | |
| **Edges** | 65,854 | 186,100 | 13,584 | 65,718 | 30,019 | 479,476 | 343,486 | 431,206 | 88,651 | 238,162 | |
| **Classes** | 18 | 5 | 5 | 5 | 5 | 9 | 6 | 10 | 3 | 8 | |
| MLP | 62.29 ± 1.03 | 42.66 ± 0.84 | 38.66 ± 4.02 | 36.74 ± 1.80 | 36.70 ± 0.85 | 89.82 ± 0.63 | 93.57 ± 0.55 | 78.94 ± 1.22 | 87.48 ± 0.46 | 89.96 ± 1.22 | 13.7 |
| GCN | 38.58 ± 2.35 | 45.16 ± 0.49 | 42.12 ± 3.82 | 38.47 ± 1.82 | 30.11 ± 0.74 | 68.25 ± 2.75 | 78.15 ± 0.95 | 77.53 ± 1.41 | 87.70 ± 0.32 | 94.31 ± 0.33 | 13.6 |
| GAT | 59.55 ± 1.45 | 47.72 ± 0.73 | 40.89 ± 3.50 | 38.22 ± 1.71 | 30.94 ± 0.95 | 57.22 ± 3.04 | 88.36 ± 1.37 | 76.69 ± 0.87 | 87.45 ± 0.53 | 94.59 ± 0.48 | 13.7 |
| APPNP | 70.86 ± 0.69 | 46.06 ± 0.66 | 42.18 ± 4.03 | 36.22 ± 1.54 | 35.06 ± 1.22 | 91.50 ± 0.51 | 96.29 ± 0.41 | 84.33 ± 0.73 | 89.25 ± 0.53 | 95.38 ± 0.36 | 8.4 |
| GCNII | 82.53 ± 0.37 | 47.53 ± 0.72 | 41.56 ± 4.15 | 40.70 ± 1.80 | **37.51 ± 0.92** | 91.64 ± 0.67 | 96.48 ± 0.62 | 84.63 ± 0.66 | 89.96 ± 0.43 | 95.18 ± 0.39 | 4.7 |
| H2GCN | 68.61 ± 1.05 | 37.20 ± 0.67 | 42.29 ± 4.57 | 35.82 ± 2.20 | 33.32 ± 0.90 | 91.25 ± 0.58 | 96.24 ± 0.39 | 78.34 ± 2.01 | 89.32 ± 0.37 | 95.66 ± 0.26 | 10.0 |
| MixHop | 79.16 ± 0.70 | 47.95 ± 0.65 | 44.97 ± 3.12 | 40.43 ± 1.40 | 36.97 ± 0.90 | 91.10 ± 0.46 | 96.21 ± 0.42 | 84.19 ± 0.61 | 89.42 ± 0.37 | 95.63 ± 0.30 | 5.1 |
| GBK-GNN | 66.05 ± 1.44 | 40.20 ± 1.96 | 42.01 ± 4.89 | 36.52 ± 1.45 | 35.70 ± 1.12 | OOM | OOM | 81.07 ± 0.83 | 88.18 ± 0.45 | 93.48 ± 0.42 | 13.7 |
| GGCN | OOM | OOM | 41.23 ± 4.08 | 36.76 ± 2.19 | 35.68 ± 0.87 | 90.84 ± 0.65 | 95.58 ± 0.44 | 84.76 ± 0.65 | 89.04 ± 0.40 | 95.18 ± 0.44 | 11.3 |
| GloGNN | 68.63 ± 0.63 | 48.62 ± 0.59 | 40.95 ± 5.95 | 36.85 ± 1.97 | 36.66 ± 0.81 | 90.47 ± 0.77 | 94.51 ± 0.49 | 82.83 ± 0.52 | 89.60 ± 0.34 | 95.09 ± 0.46 | 9.5 |
| HOGGCN | OOM | OOM | 43.35 ± 3.66 | 38.63 ± 1.95 | 36.47 ± 0.83 | 90.94 ± 0.72 | 94.75 ± 0.65 | 83.74 ± 0.69 | OOM | 94.79 ± 0.26 | 10.9 |
| GPR-GNN | 71.19 ± 0.75 | 46.64 ± 0.52 | 41.84 ± 4.68 | 38.04 ± 1.98 | 36.21 ± 0.98 | 91.19 ± 0.47 | 96.37 ± 0.44 | 84.07 ± 0.54 | 89.28 ± 0.37 | 95.48 ± 0.24 | 7.5 |
| ACM-GCN | 71.15 ± 0.73 | 50.64 ± 0.61 | 45.20 ± 4.14 | 40.90 ± 1.74 | 35.88 ± 1.40 | 91.43 ± 0.65 | 96.19 ± 0.45 | 84.39 ± 0.43 | 89.99 ± 0.40 | 95.52 ± 0.40 | 4.6 |
| OrderedGNN | 83.10 ± 0.75 | 51.30 ± 0.61 | 42.07 ± 4.24 | 37.75 ± 2.53 | 37.22 ± 0.62 | 91.42 ± 0.79 | 96.27 ± 0.73 | 85.50 ± 0.80 | 90.09 ± 0.37 | 95.73 ± 0.33 | 3.8 |
| CLP | 67.36 ± 0.54 | 47.42 ± 0.44 | 41.96 ± 4.18 | 37.75 ± 1.37 | 35.34 ± 0.74 | 90.20 ± 0.64 | 94.46 ± 0.58 | 83.17 ± 0.86 | 88.92 ± 0.32 | 93.52 ± 0.57 | 11.0 |
| EPFGNN | 43.11 ± 0.78 | 45.31 ± 0.63 | 44.08 ± 4.57 | 41.10 ± 2.52 | 30.03 ± 1.22 | 57.91 ± 2.23 | 74.29 ± 3.24 | 80.98 ± 0.57 | 87.07 ± 0.53 | 91.08 ± 0.58 | 13.1 |
| CPGNN | 59.55 ± 0.84 | 46.65 ± 0.71 | 41.45 ± 4.84 | 37.24 ± 2.09 | 33.37 ± 1.02 | 80.46 ± 1.25 | 81.92 ± 1.06 | 77.87 ± 1.65 | 87.98 ± 0.40 | 93.35 ± 0.58 | 13.6 |
| **CMGNN** | **84.35 ± 1.27** | **52.13 ± 0.55** | **45.70 ± 4.92** | **41.89 ± 2.34** | 36.82 ± 0.78 | **92.66 ± 0.46** | **97.00 ± 0.52** | 84.50 ± 0.73 | 89.99 ± 0.32 | 95.48 ± 0.29 | **2.1** |

**Objective Function.** As mentioned in Sec 4, the CMs in real-world graphs don't always have significant discriminability, which may lead to low effectiveness of supplementary messages. Thus, we introduce an additional discrimination loss $\mathcal{L}_{dis}$ to reduce the similarity of the desired neighborhood message among different classes, which enhances the discriminability among classes in CM. The overall loss consists of a CrossEntropy loss $\mathcal{L}_{ce}$ and the discrimination loss $\mathcal{L}_{dis}$:

$$\mathcal{L} = \mathcal{L}_{ce}(\tilde{\mathbf{Z}}, \mathbf{Y}) + \lambda \mathcal{L}_{dis}, \quad \mathcal{L}_{dis} = \sum_{i \neq j} \text{Sim}(\hat{\mathbf{M}}_i \mathbf{Z}_{ptt}, \hat{\mathbf{M}}_j \mathbf{Z}_{ptt}), \tag{15}$$

where $\mathbf{Z}_{ptt} \in \mathbb{R}^{K \times d_r}$ is the representation of virtual prototypes nodes. More details of CMGNN including pseudo code are available in Appendix F.

## 6 BENCHMARKS AND EXPERIMENTS

In this section, we conduct comprehensive experiments to demonstrate the effectiveness of the proposed CMGNN with a newly organized benchmark for fair comparisons.

### 6.1 NEW BENCHMARK

As reported in Platonov et al. (2023), some widely adopted datasets in existing works have critical drawbacks, which lead to unreliable results. Therefore, with a comprehensive review of existing benchmark evaluation, we construct a new benchmark to fairly perform experimental validation. Specifically, we integrate 17 representative homophilous and heterophilous GNNs, construct a unified codebase, and evaluate their node classification performances on 10 unified organized datasets with various heterophily levels.

**Drawbacks of Existing Datasets.** Existing works mostly follow the settings and datasets used in Pei et al. (2020), including 6 heterophilous datasets (Cornell, Texas, Wisconsin, Actor, Chameleon, and Squirrel) and 3 homophilous datasets (Cora, Citeseer, and Pubmed). Platonov et al. (2023) pointed out serious data leakages in Chameleon and Squirrel, while Cornell, Texas, and Wisconsin are too small with very imbalanced classes. Further, we revisit other datasets and discover new drawbacks: (i) In the ten splits of Citeseer, there are two inconsistent ones, which have smaller training, validation, and test sets that could cause issues with statistical results; (ii) Cora's data split ratios are inconsistent with the expected ones. These drawbacks may lead to certain issues in the conclusions of previous works. The details of dataset drawbacks are listed in Appendix G.1.

**Newly Organized Datasets.** To avoid the issues of method comparison caused by above drawbacks, we have collected and filtered suitable graph datasets from heterophilous GNNs methods and other fields (e.g. Anomaly Detection). This collection spans various levels of homophily, providing a robust

foundation for performance evaluation. The datasets used in the benchmark include Roman-Empire, Amazon-Ratings, Chameleon-F, Squirrel-F, Actor, Flickr, BlogCatalog, Wikics, Pubmed, and Photo. Their statistics are summarized in Table 2, with details in Appendix G.2. For consistency with existing methods, we randomly construct 10 splits with predefined proportions (48%/32%/20% for train/valid/test) for each dataset and report the mean performance and standard deviation of 10 splits.

**Baseline Methods.** As baseline methods, we choose 17 representative homophilous and heterophilous GNNs, including (i) shallow base model: MLP; (ii) homophilous GNNs: GCN (Kipf & Welling, 2017), GAT (Veličković et al., 2018), APPNP (Gasteiger et al., 2019), GCNII (Chen et al., 2020); (iii) heterophilous GNNs: H2GCN (Zhu et al., 2020), MixHop (Abu-El-Haija et al., 2019), GBK-GNN (Du et al., 2022), GGCN (Yan et al., 2022), GloGNN (Li et al., 2022), HOGGCN (Wang et al., 2022), GPR-GNN (Chien et al., 2021), ACM-GCN (Luan et al., 2022) and OrderedGNN (Song et al., 2023), (iv) compatibility matrix based methods: CLP (Zhong et al., 2022), EPFGNN (Wang et al., 2021), CPGNN (Zhu et al., 2021a). For each method, we integrate its official/reproduced code into a unified codebase and search for parameters in the space suggested by the original papers. All methods share the same call interfaces, ensuring a fair comparison environment. More experimental settings can be found in Appendix G.4 and H.1.

## 6.2 MAIN RESULTS

Following the constructed benchmark, we evaluate methods and report the performance in Table 2.

**Performance of Baseline Methods.** With the new benchmarks, some interesting observations and conclusions can be found when analyzing the performance of baseline methods. First, comparing the performance of MLP and GCN, we can find "good" heterophily in Amazon-Ratings, Chameleon-F, and Squirrel-F, where GCN performs better than MLP under this kind of heterophily. Meanwhile, "bad" homophily may also exist as shown in BlogCatalog and Wikics, where the homophily level is insufficient for vanilla message-passing methods (GCN, GAT) to outperform MLP. These results once again support the observations about CMs. Therefore, **homophilous GNNs** can also work well in heterophilous graphs as GCNII has an average rank of 4.7, which is better than most HTGNNs. This is attributed to the initial residual connection in GCNII actually playing the role of ego/neighbor separation, which is suitable in heterophilous graphs. As for **heterophilous GNNs**, they are usually designed for both homophilous and heterophilous graphs. Surprisingly, MixHop, as an early method, demonstrated quite good performance. In fact, from the perspective of HTMP, it can be considered a degenerate version of OrderedGNN with no learnable dimensions. As previous SOTA methods, OrderedGNN and ACM-GCN prove their strong capabilities again.

**Performance of CMGNN.** CMGNN achieves the best performance in 6 datasets and an average rank of 2.1, which outperforms baseline methods. This demonstrates the superiority of utilizing and enhancing the CM to handle incomplete and noisy semantic neighborhoods, especially in heterophilous graphs. Regarding the suboptimal performance in Actor, we believe that this is due to the CM in this dataset are not discriminative enough to provide valuable information via the supplementary messages and hard to enhance. In homophilous graphs, due to the identity-like CMs, the overlap between distributions is relatively less, leading to a minor contribution from supplement messages. Yet CMGNN still achieves top-level performances.

**Comparision with CM-based methods.** Some existing methods also utilize the compatibility matrix (CM) to redefine pair-wise relations (i.e. edge weights) for existing edges, such as label propagation in CLP, log-likelihood estimation in EPFGNN, and prior belief propagation in CPGNN. In contrast, CMGNN leverages CM and virtual neighbors to construct supplementary messages while preserving the original neighborhood distribution. As a result, CMGNN achieves better performances and benefits from the approach of utilizing CM in the following aspects: (i) Better robustness for low-quality pseudo labels; (ii) Unlock the effectiveness of CM for low-degree nodes; (iii) More accurate estimation of CM. More detailed analyses are available in Appendix H.2.1

## 6.3 ABLATION STUDY

We conduct an ablation study on two key designs of CMGNN , including the supplementary messages of the desired neighborhood (SM) and the discrimination loss (DL). The results are shown in Table 3. *First of all*, both SM and DL have indispensable contributions except for Flickr, BlogCatalog, and

Table 3: Ablation study results (%) between CMGNN and three ablation variants, where SM denotes supplementary messages of the desired neighborhoods and DL denotes the discrimination loss.

| Variants | Roman-Empire | Amazon-Ratings | Chameleon-F | Squirrel-F | Actor | Flickr | BlogCatalog | Wikics | Pubmed | Photo |
|---|---|---|---|---|---|---|---|---|---|---|
| **CMGNN** | **84.35 ± 1.27** | **52.13 ± 0.55** | **45.70 ± 4.92** | **41.89 ± 2.34** | **36.82 ± 0.78** | **92.66 ± 0.46** | **97.00 ± 0.52** | **84.50 ± 0.73** | **89.99 ± 0.32** | **95.48 ± 0.29** |
| W/O SM | 83.84 ± 1.09 | 51.98 ± 0.61 | 42.35 ± 4.21 | 40.79 ± 1.89 | 36.02 ± 1.21 | 92.32 ± 0.83 | 96.52 ± 0.63 | 83.97 ± 0.83 | 89.70 ± 0.44 | 95.41 ± 0.40 |
| W/O DL | 83.68 ± 1.24 | 52.04 ± 0.37 | 44.97 ± 3.99 | 41.60 ± 2.43 | 36.28 ± 1.12 | **92.66 ± 0.46** | **97.00 ± 0.52** | 83.29 ± 1.83 | **89.99 ± 0.32** | 95.26 ± 0.35 |
| W/O SM and DL | 83.52 ± 1.91 | 51.58 ± 1.04 | 41.12 ± 2.93 | 40.07 ± 2.41 | 35.61 ± 1.48 | 92.32 ± 0.83 | 96.52 ± 0.63 | 81.62 ± 1.67 | 89.70 ± 0.44 | 94.66 ± 0.42 |

Table 4: Node classification accuracy (%) comparison among nodes with different degrees.

| Dataset | Amazon-Ratings | | | | | Flickr | | | | | BlogCatalog | | | | |
|---|---|---|---|---|---|---|---|---|---|---|---|---|---|---|---|
| Deg. Prop.(%) | 0~20 | 20~40 | 40~60 | 60~80 | 80~100 | 0~20 | 20~40 | 40~60 | 60~80 | 80~100 | 0~20 | 20~40 | 40~60 | 60~80 | 80~100 |
| **CMGNN** | **59.78** | **58.36** | **53.08** | 41.74 | 47.86 | **92.56** | **91.19** | 92.71 | **93.24** | 93.65 | **94.13** | **97.17** | **98.29** | **97.99** | **97.47** |
| ACM-GCN | 57.35 | 56.21 | 51.74 | 41.55 | 46.47 | 90.44 | 91.17 | **92.85** | 93.19 | 89.50 | 92.17 | 96.68 | 97.83 | 97.84 | 96.51 |
| OrderedGNN | 56.32 | 56.16 | 51.20 | **41.85** | **50.26** | 86.48 | 90.07 | 92.40 | 92.79 | 93.40 | 92.19 | 96.09 | 97.48 | 97.36 | 96.27 |
| GCNII | 50.61 | 49.94 | 47.49 | **41.85** | 47.76 | 87.49 | 90.54 | 92.29 | 92.68 | **95.09** | 92.81 | 96.73 | 97.58 | 97.90 | 97.43 |

Pubmed, in which the discrimination loss has no effect. Specifically, the best choice of parameter $\lambda$ on these datasets is 0 thus resulting in the identical performance in both "CMGNN" and "W/O DL" settings. This may be due to the discriminability of desired neighborhood messages reaching the bottlenecks and can not be further improved by DL. *Meanwhile*, the extent of their contributions varies across datasets. SM plays a more important role in most datasets except Roman-Empire, Wikics, and Photo, in which the number of nodes that need supplementary messages is relatively small and DL has great effects. *Further*, we notice that with SM and DL, CMGNN can reach a smaller standard deviation most of the time. This illustrates that CMGNN achieves more stable results by handling nodes with incomplete and noisy semantic neighborhoods. As for the opposite result on Chameleon-F, this may attributed to the small size of this dataset (890 nodes), which can lead to naturally unstable results.

## 6.4 Performance on Nodes with Various Levels of Degrees

To verify the effect of CMGNN on nodes with incomplete and noisy semantic neighborhoods, we divide the test set nodes into 5 parts according to their degrees and report the classification accuracy respectively. We compare CMGNN with 3 top-performance methods and show the results in Table 4. In general, nodes with low degrees tend to have incomplete and noisy semantic neighborhoods. Thus, our outstanding performances on the top 20% nodes with the least degree demonstrate the effectiveness of CMGNN for providing desired neighborhood messages. Further, we can find that OrderedGNN and GCNII are good at dealing with nodes with high degrees, while ACM-GCN is relatively good at nodes with low degrees. And CMGNN , to a certain extent, can be adapted to both situations at the same time.

More detailed experimental results can be found in Appendix H.2, such as more ablation studies, scalability studies on large-scale graphs, comprehensive complexity analysis and comparison.

## 7 Conclusion and Limitations

In this paper, we revisit the message-passing mechanism in existing heterophilous GNNs and reformulate them into a unified heterophilous message-passing (HTMP) mechanism. Based on the HTMP mechanism and empirical analysis, we reveal that the reason for message passing remaining effective is attributed to implicitly enhancing the compatibility matrix among classes. Further, we propose a novel method CMGNN to unlock the potential of the compatibility matrix by handling the incomplete and noisy semantic neighborhoods. The experimental results show the effectiveness of CMGNN and the feasibility of designing a new method following HTMP mechanism. We hope the HTMP mechanism and benchmark can further provide convenience to the community.

This work mainly focuses on the message-passing mechanism in existing HTGNNs under the semi-supervised setting. Thus, the other designs in HTGNNs such as objective functions are not analyzed in this paper. The proposed HTMP mechanism is suitable for only a large part of existing HTGNNs which still follow the message passing mechanism.

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

## A  RELATED WORKS

**Homophilous Graph Neural Networks**. Graph Neural Networks (GNNs) have showcased impressive capabilities in handling graph-structured data (Wu et al., 2020; Chen et al., 2024). Traditional GNNs are predominantly founded on the assumption of homophily, broadly categorized into two classes: spectral-based GNNs and spatial-based GNNs. **Firstly**, spectral-based GNNs acquire node

representations through graph convolution operations employing diverse graph filters (Kipf & Welling, 2017; Defferrard et al., 2016; Xu et al., 2018a). **Secondly**, spatial-based methods gather information from neighbors and update the representation of central nodes through the message-passing mechanism (Gasteiger et al., 2019; Veličković et al., 2018; Hamilton et al., 2017). Moreover, for a more **comprehensive understanding of existing homophilous GNNs**, several unified frameworks (Ma et al., 2021; Zhu et al., 2021b) have been proposed. Ma et al. (2021) propose that the aggregation process in some representative homophilous GNNs can be regarded as solving a graph denoising problem with a smoothness assumption. Zhu et al. (2021b) establishes a connection between various message-passing mechanisms and a unified optimization problem. However, these methods have limitations, as the aggregated representations may lose discriminability when heterophilous neighbors dominate (Bo et al., 2021; Zhu et al., 2020).

**Heterophilous Graph Neural Networks**. Recently, some heterophilous GNNs have emerged to tackle the heterophily problem (Bo et al., 2021; Zhu et al., 2020; Jin et al., 2021a;b; Pei et al., 2020; Abu-El-Haija et al., 2019; Wang et al., 2022; Luan et al., 2022; Li et al., 2022; Chien et al., 2021; Song et al., 2023; Suresh et al., 2021; Yan et al., 2022). **Firstly**, a commonly adopted strategy involves *expanding the neighborhood with higher homophily or richer messages*, such as high order neighborhooods (Zhu et al., 2020; Jin et al., 2021a), feature-similarity-based neighborhoods (Jin et al., 2021a;b), and custom-defined neighborhoods (Pei et al., 2020; Suresh et al., 2021). **Secondly**, some approaches (Bo et al., 2021; Wang et al., 2022; Luan et al., 2022; Li et al., 2022; Yan et al., 2022) aim to *leverage information from heterophilous neighbors*, considering that not all heterophily is detrimental et al.(Ma et al., 2022). **Thirdly**, some methods (Zhu et al., 2020; Abu-El-Haija et al., 2019; Chien et al., 2021; Song et al., 2023) adapt to heterophily by extending the combine function in message passing, creating variations for addition and concatenation.

**Reviewing Heterophilous Graph Neural Networks**. Due to heterophilous GNNs have attracted more and more research attention. Some surveys have provided a macroscopic view for reviewing heterophilous GNNs, categorizing heterophilous GNNs with shallow analysis. Specifically, Zheng et al. (2022) categorizes the designs of heterophilous GNNs into non-local neighbor extensions and GNN architecture refinement. Zhu et al. (2023) examines the impact of heterophilous graph characteristics on GNNs. For categorizations, it simply lists some effective designs in heterophilous GNNs. Gong et al. (2024) reviews heterophilous graph learning, where message passing is only a minor aspect of its taxonomy with a broader view. However, these works offer guidance primarily at the conceptual level and categorize existing heterophilous GNNs based on literature summaries, lacking in-depth analysis of message-passing mechanisms. In this paper, we provide a uniform symbolic form and categorize existing methods based on the values of component modules. Further, our review guides the design of new SOTA heterophilous message-passing mechanisms.

# B  MORE DETAILS OF HTMP MECHANISM

In this part, we list more details about the HTMP mechanism, including additional analysis about HTMP, method-wise analysis, and overall analysis.

## B.1  ADDITIONAL ANALYSIS OF HTMP MECHANISM

### B.1.1  NEIGHBORHOOD INDICATORS

The neighborhood indicator explicitly marks the neighbors of all nodes within a specific neighborhood. In existing heterophilous GNNs, neighborhood indicators typically take one of the following forms: (i) Raw Neighborhood (Raw); (ii) Neighborhood Redefining (ReDef); and (3) Neighborhood Discrimination (Dis).

**Raw Neighborhood.** Raw neighborhood, including $\mathbf{A}$ and $\tilde{\mathbf{A}}$, provides the basic neighborhood information. The only difference between them is whether there is differential treatment of the node's ego messages. For example, APPNP (Gasteiger et al., 2019) applies additional weighting to the ego messages of nodes compared to GCN (Kipf & Welling, 2017). In heterophilous GNNs, ego/neighbor separation is a common strategy that can mitigate the confusion of ego messages with neighbor messages.

**Neighborhood Redefining.** Neighborhood redefining is the most commonly used technique in heterophilous GNNs, aiming to capture additional information from new neighborhoods. As a representative example, *high-order neighborhood* $\mathbf{A}_h$ can provide long-distance connection information but also result in additional computational costs. *Feature-similarity-based neighborhood* $\mathbf{A}_f$ is often defined by the k-NN relationships within the feature space. Fundamentally, it only utilizes node features and thus needs to be used in conjunction with other neighborhood indicators. Otherwise, the model will be limited by the amount of information in node features. GloGNN (Li et al., 2022) introduces *fully-connected neighborhood* $\mathbf{1} \in \mathbb{R}^{N \times N}$, which can capture global neighbor information from all nodes. However, it can also cause significant time and space consumption. Additionally, there are some *custom-defined neighborhood* $\mathbf{A}_c$. For example, Geom-GCN (Pei et al., 2020) redefines neighborhoods based on the geometric relationships between node pairs. These neighborhood indicators may have limited generality, and the effectiveness is reliant on the specific method.

**Neighborhood Discrimination.** Neighborhood discrimination aims to mark whether neighbors share the same label with central nodes. The neighborhoods are partitioned into positive $\mathbf{A}_p$ and negative ones $\mathbf{A}_n$, which include homophilous and heterophilous neighbors respectively. GGCN (Yan et al., 2022) divides the raw neighborhood based on the similarity of node representations with a threshold of 0. Explicitly distinguishing neighbors allows for targeted processing, making the model more interpretable. However, its performance is influenced by the accuracy of the discrimination, which may lead to the accumulation of errors.

### B.1.2 AGGREGATION GUIDANCE

After identifying the neighborhood, the aggregation guidance controls what type of messages to gather from the corresponding neighbors. The existing aggregation guidance mainly includes three kinds of approaches: (1) Degree Averaging (DegAvg), (2) Adaptive Weights (AdaWeight), and (3) Relationship Estimation (RelaEst).

**Degree Averaging.** Degree averaging, formatted as $\mathbf{B}^d = \mathbf{D}^{-\frac{1}{2}}\mathbf{1}\mathbf{D}^{-\frac{1}{2}}$ or $\mathbf{B}^d = \mathbf{D}^{-1}\mathbf{1}$, is the most common aggregation guidance, which plays the role of a low-pass filter to capture the smooth signals and is fixed during model training. Further, combining negative degree averaging with an identity aggregation guidance $\mathbf{I} \in \mathbb{R}^{N \times N}$ can capture the difference between central nodes and neighbors, as used in ACM-GCN (Luan et al., 2022). Degree averaging is simple and efficient but depends on the discriminability of corresponding neighborhoods.

**Adaptive Weights.** Another common strategy is allowing the model to learn the appropriate aggregation guidances $\mathbf{B}^{aw}$. GAT (Veličković et al., 2018) proposes an attention mechanism to learn aggregate weights, which guides many subsequent heterophilous methods. To better handle heterophilous graphs, FAGCN (Bo et al., 2021) introduces negative-available attention weights $\mathbf{B}^{naw}$ to capture the difference between central nodes and heterophilous neighbors. Adaptive weights can personalize message aggregation for different neighbors, yet it's difficult for models to attain the desired effect.

**Relationship Estimation.** Recently, some methods have tried to estimate the pair-wise relationships $\mathbf{B}^{re}$ between nodes and use them to guide message aggregation. HOG-GCN (Wang et al., 2022) estimates the pair-wise homophily levels between nodes as aggregation guidances based on both attribute and topology space. GloGNN (Li et al., 2022) treats all nodes as neighbors and estimates a coefficient matrix as aggregation guidance based on the idea of linear subspace expression. GGCN (Yan et al., 2022) estimates appropriate weights for message aggregation with the degrees of nodes and the similarities between node representations. Relationship estimation usually has theoretical guidance, which brings strong interpretability. However, it may also result in significant temporal and spatial complexity when estimating pair-wise relations.

### B.1.3 COMBINE FUNCTION

After message aggregation, the COMBINE functions integrate messages from multiple neighborhoods into layer representations. COMBINE functions in heterophilous GNNs are commonly based on two operations: addition and concatenation, each of which has variants. To merge several messages together, addition (Add) is a naive idea. Further, to control the weight of messages from different neighborhoods, weighted addition (WeightedAdd) is applied. However, it is a global setting and cannot adapt to the differences between nodes. Thus, adaptive weighted addition (AdaAdd) is

proposed, which can learn personalized message combination weights for each node, but it will result in additional time consumption. Although the addition is simple and efficient, some methods (Zhu et al., 2020; Abu-El-Haija et al., 2019) believe that it may blur messages from different neighborhoods, which can be harmful in heterophilous GNNs, so they employ a concatenation operation (Cat) to separate the messages. Nevertheless, such an approach not only increases the space cost but may also retain additional redundant messages. To address these issues, OrderedGNN (Song et al., 2023) proposes an adaptive concatenation mechanism (AdaCat) that can combine multiple messages with learnable dimensions. This is an innovative and worthy further exploration practice, but the difficulty of model learning should also be considered.

### B.1.4 FUSE FUNCTION

Further, the FUSE functions integrate messages from multiple layers into the final representation. For the FUSE function, utilizing the representation of the last layer as the final representation is widely accepted: $\mathbf{Z} = \mathbf{Z}^L$. JKNet (Xu et al., 2018b) proposes that the combination of representations from intermediate layers can capture both local and global information. H2GCN (Zhu et al., 2020) applies it in heterophilous graphs, preserving messages from different localities with concatenation. Similarly, GPRGNN (Chien et al., 2021) combines the representations of multiple layers into the final representation through adaptive weighted addition.

### B.1.5 AGGREGATE FUNCTION

The most commonly used AGGREGATE function is $\mathbf{AGGREGATE}(\mathbf{A}_r, \mathbf{B}_r, \mathbf{Z}_r^{l-1}) = (\mathbf{A}_r \odot \mathbf{B}_r)\mathbf{Z}_r^{l-1}\mathbf{W}_r^l$. We take this as the fixed form of the AGGREGATE function following. Actually, the input representations $\mathbf{Z}_r^{-1}$ and weight matrixes $\mathbf{W}_r^l$ also can be specially designed. Taking the initial node representations $\mathbf{Z}^0$ as input is a relatively common approach as in APPNP (Gasteiger et al., 2019), GCNII (Chen et al., 2020), FAGCN (Bo et al., 2021) and GloGNN (Li et al., 2022). Further, GCNII (Chen et al., 2020) adds an identity matrix $\mathbf{I}_w$ to the weight matrixes to keep more original messages. However, the methods that specially design these components are few and with a similar form. Thus, we don't discuss them too much, but leave it for future extensions.

### B.2 REVISITING REPRESENTATIVE GNNs WITH HTMP MECHANISM

In this part, we utilize HTMP mechanism to revisit the representative GNNs. We start from homophilous GNNs as simple examples and further extend to heterophilous GNNs.

### B.2.1 GCN

Graph Convolutional Networks (GCN) (Kipf & Welling, 2017) utilizes a low-pass filter to gather messages from neighbors as follows:

$$\mathbf{Z}^l = \hat{\tilde{\mathbf{A}}}\mathbf{Z}^{l-1}\mathbf{W}^l. \tag{16}$$

It can be revisited by HTMP with the following components:

$$\mathbf{A}_0 = \tilde{\mathbf{A}}, \quad \mathbf{B}_0 = \mathbf{B}^d = \tilde{\mathbf{D}}^{-\frac{1}{2}}\mathbf{1}\tilde{\mathbf{D}}^{-\frac{1}{2}},$$
$$\mathbf{Z}^l = \mathbf{Z}_0^l = (\mathbf{A}_0 \odot \mathbf{B}_0)\mathbf{Z}^{l-1}\mathbf{W}^l = \hat{\tilde{\mathbf{A}}}\mathbf{Z}^{l-1}\mathbf{W}^l. \tag{17}$$

Specifically, GCN has a raw neighborhood indicator $\tilde{\mathbf{A}}$ and a degree averaging aggregation guidance $\mathbf{B}^d$. Since there is only one neighborhood, the COMBINE function is meaningless in GCN. GCN utilizes a naive way to fuse messages about the original neighborhood and central nodes. However, it may confuse the representations in heterophilous graphs.

### B.2.2 APPNP

PPNP (Gasteiger et al., 2019) is also a general method whose message passing is based on Personalized PageRank (PPR). To avoid massive consumption, APPNP is introduced as the approximate version of PPNP with an iterative message-passing mechanism:

$$\mathbf{Z}^l = \mu\mathbf{Z}^0 + (1 - \mu)\hat{\mathbf{A}}\mathbf{Z}^{l-1}. \tag{18}$$

It can be revisited by _with the following components:

$$\mathcal{A} = [\mathbf{A}_0, \ \mathbf{A}_1], \quad \mathcal{B} = [\mathbf{B}_0, \ \mathbf{B}_1],$$
$$\mathbf{A}_0 = \mathbf{I}, \quad \mathbf{B}_0 = \mathbf{I}, \quad \mathbf{W}_0^l = \mathbf{I},$$
$$\widetilde{\mathbf{Z}}_0^l = (\mathbf{A}_0 \odot \mathbf{B}_0)\mathbf{Z}^0\mathbf{W}_0^l = \mathbf{Z}^0, \tag{19}$$
$$\mathbf{A}_1 = \mathbf{A}, \quad \mathbf{B}_1 = \mathbf{D}^{-\frac{1}{2}}\mathbf{1}\mathbf{D}^{-\frac{1}{2}}, \quad \mathbf{W}_1^l = \mathbf{I},$$
$$\widetilde{\mathbf{Z}}_1^l = (\mathbf{A}_1 \odot \mathbf{B}_1)\mathbf{Z}^{l-1}\mathbf{W}_1^l = \hat{\mathbf{A}}\mathbf{Z}^{l-1}.$$

Specifically, APPNP aggregates messages from node ego and neighborhoods separately and combines them with a weighted addition. Compared with GCN, APPNP assigns adjustable weights to nodes, for controlling the proportion of ego and neighbor messages during message-passing, which becomes a worthy design in heterophilous graphs.

### B.2.3 GAT

Going a step further, Graph Attention Networks (GAT) (Veličković et al., 2018) allows learnable weights for each neighbor:

$$\mathbf{Z}_i^l = \sum_{j \in \tilde{\mathcal{N}}(i)} \alpha_{ij}\mathbf{Z}_j^{l-1}\mathbf{W}^l, \tag{20}$$

where $\alpha_{ij}$ is the weight for aggregating neighbor node $j$ to center node $i$, whose construction process is as follows:

$$\alpha_{ij} = \frac{\exp(e_{ij})}{\sum_{k \in \tilde{\mathcal{N}}(i)} \exp(e_{ik})}, \tag{21}$$
$$e_{ij} = \text{LeakyReLU}\left(\left[\mathbf{Z}_i^{l-1}|\mathbf{Z}_j^{l-1}\right]\mathbf{a}\right).$$

Let $\mathbf{P}^{GAT}$ be the matrix of aggregation weights in GAT:

$$\mathbf{P}_{ij}^{GAT} = \begin{cases} \alpha_{ij}, & \tilde{\mathbf{A}}_{ij} = 1, \\ 0, & \tilde{\mathbf{A}}_{ij} = 0. \end{cases} \tag{22}$$

HTMP can revisit GAT with the following components:

$$\mathbf{A}_0 = \tilde{\mathbf{A}}, \quad \mathbf{B}_0 = \mathbf{B}^{aw} = \mathbf{P}^{GAT},$$
$$\mathbf{Z}^l = \mathbf{Z}_0^l = (\mathbf{A}_0 \odot \mathbf{B}_0)\mathbf{Z}^{l-1}\mathbf{W}^l = \mathbf{P}^{GAT}\mathbf{Z}^{l-1}\mathbf{W}^l, \tag{23}$$

which is the matrix version of Eq 20. Specifically, GAT aggregate messages from raw neighborhood $\tilde{\mathbf{A}}$ with adaptive weights $\mathbf{B}^{aw}$. Aggregation guidance with adaptive weights is a nice idea, but simple constraints are not enough for the model to learn ideal results.

### B.2.4 GCNII

GCNII (Chen et al., 2020) is a novel homophilous GNN with two key designs: initial residual connection and identity mapping, which can be formatted as follows:

$$\mathbf{Z}^l = \left(\alpha\mathbf{Z}^0 + (1-\alpha)\tilde{\mathbf{D}}^{-\frac{1}{2}}\tilde{\mathbf{A}}\tilde{\mathbf{D}}^{-\frac{1}{2}}\mathbf{Z}^{l-1}\right)\left(\beta\mathbf{W}^l + (1-\beta)\mathbf{I}_w\right), \tag{24}$$

where $\alpha$ and $\beta$ are two predefined parameters and $\mathbf{I}_w \in \mathbb{R}^{d_r \times d_r}$ is an identity matrix.

From the perspective of HTMP, it can be viewed as follows:

$$\mathcal{A} = [\mathbf{I}, \tilde{\mathbf{A}}], \quad \mathcal{B} = [\mathbf{I}, \tilde{\mathbf{B}}^d], \quad \mathbf{W}_0^l = \mathbf{W}_1^l = \left(\beta\mathbf{W}^l + (1-\beta)\mathbf{I}_w\right),$$
$$\widetilde{\mathbf{Z}}_0^l = (\mathbf{I} \odot \mathbf{I})\mathbf{Z}^0\left(\beta\mathbf{W}^l + (1-\beta)\mathbf{I}_w\right) = \mathbf{Z}^0\left(\beta\mathbf{W}^l + (1-\beta)\mathbf{I}_w\right), \tag{25}$$
$$\widetilde{\mathbf{Z}}_1^l = (\tilde{\mathbf{A}} \odot \tilde{\mathbf{B}}^d)\mathbf{Z}^{l-1}\left(\beta\mathbf{W}^l + (1-\beta)\mathbf{I}_w\right) = \hat{\mathbf{A}}\mathbf{Z}^{l-1}\left(\beta\mathbf{W}^l + (1-\beta)\mathbf{I}_w\right),$$

where the COMBINE function is weighted addition. Specifically, the first design of GCNII is a form of ego/neighbor separation, and the second design is a novel transformation weights matrix. This can also be specially designed, but only GCNII does this, so we won't analyze it too much and leave it as a future extension.

### B.2.5 GEOM-GCN

Geom-GCN (Pei et al., 2020) is one of the most influential heterophilous GNNs, which employs the geometric relationships of nodes within two kinds of neighborhoods to aggregate the messages through bi-level aggregation:

$$\mathbf{Z}^l = \left( \underset{i\in\{g,s\}}{\|} \underset{r\in R}{\|} \mathbf{Z}^l_{i,r} \right) \mathbf{W}^l,$$

$$\mathbf{Z}^l_{i,r} = \mathbf{D}^{-\frac{1}{2}}_{i,r} \mathbf{A}_{i,r} \mathbf{D}^{-\frac{1}{2}}_{i,r} \mathbf{Z}^{l-1}, \tag{26}$$

where $\|$ denotes the concatenate operator, $\{g, s\}$ is the set of neighborhoods including the original graph and the latent space. $R$ is the set of geometric relationships. $\mathbf{A}_{i,r}$ is the corresponding adjacency matrix in neighborhood $i$ and relationship $r$.

It can be revisited by HTMP with the following components:

$$\mathcal{A} = [\mathbf{A}_{i,r}|i \in \{g,s\}, r \in R], \quad \mathcal{B} = [\mathbf{B}^d_{i,r}\|i \in \{g,s\}, r \in R],$$

$$\widetilde{\mathbf{Z}}^l_{i,r} = (\mathbf{A}_{i,r} \odot \mathbf{B}^d_{i,r})\mathbf{Z}_{l-1}\mathbf{W}^l_{i,r} = \mathbf{D}^{-\frac{1}{2}}_{i,r} \mathbf{A}_{i,r} \mathbf{D}^{-\frac{1}{2}}_{i,r} \mathbf{Z}^{l-1}\mathbf{W}^l_{i,r}, \tag{27}$$

where the COMBINE function is concatenation and the weight matrix $\mathbf{W}^l$ in Eq 26 can be viewed as the combination of multiple $\mathbf{W}^l_{i,r}$. Specifically, Geom-GCN redefines multiple neighborhoods based on the customized geometric relations in both raw and latent space. The messages are aggregated from each neighborhood and combined by a concatenation. This approach may be applicable to some datasets, yet it has weak universality.

### B.2.6 H2GCN

H2GCN (Zhu et al., 2020) is also an influential method with three key designs: ego- and neighbor-message separation, higher-order neighborhoods, and the combination of intermediate representations. Its single-layer representations are constructed as follows:

$$\mathbf{Z}^l = \left[ \hat{\mathbf{A}}\mathbf{Z}^{l-1} \| \hat{\mathbf{A}}_{h2}\mathbf{Z}^{l-1} \right], \tag{28}$$

where $\hat{\mathbf{A}}_{h2}$ denotes the 2-order adjacency matrix with normalization.

It can be revisited by HTMP with the following components:

$$\mathcal{A} = [\mathbf{A}, \mathbf{A}_{h2}], \quad \mathcal{B} = [\mathbf{B}^d, \mathbf{B}^d_{h2}], \quad \mathbf{W}^l_0 = \mathbf{W}^l_1 = \mathbf{I},$$

$$\widetilde{\mathbf{Z}}^l_0 = (\mathbf{A} \odot \mathbf{B}^d)\mathbf{Z}^{l-1}\mathbf{I} = \hat{\mathbf{A}}\mathbf{Z}^{l-1},$$

$$\widetilde{\mathbf{Z}}^l_1 = (\mathbf{A}_{h2} \odot \mathbf{B}^d_{h2})\mathbf{Z}^{l-1}\mathbf{I} = \hat{\mathbf{A}}_{h2}\mathbf{Z}^{l-1}, \tag{29}$$

where the COMBINE function is concatenation. Meanwhile, H2GCN also uses the concatenation as the FUSE function. Specifically, H2GCN aggregates messages from the raw and 2-order neighborhoods in a layer of message passing and keeps them apart in the representations. The design of ego/neighbor separation is first introduced by H2GCN and gradually becomes a necessity for subsequent methods.

### B.2.7 SIMP-GCN

SimP-GCN (Jin et al., 2021b) constructs an additional graph based on the feature similarity. It has two key concepts: (1) the information from the original graph and feature kNN graph should be balanced, and (2) each node can adjust the contribution of its node features. Specifically, the message passing in SimP-GCN is as follows:

$$\mathbf{Z}^l = \left( \text{diag}(\mathbf{s}^l)\hat{\tilde{\mathbf{A}}} + \text{diag}(1 - \mathbf{s}^l)\hat{\mathbf{A}}_f + \gamma \mathbf{D}^l_K \right) \mathbf{Z}^{l-1}\mathbf{W}^l, \tag{30}$$

where $\mathbf{s}^l \in \mathbb{R}^n$ is a learnable score vector that balances the effect of the original and feature graphs, $\mathbf{D}^l_K = \text{diag}(K^l_1, K^l_2, ..., K^l_n)$ is a learnable diagonal matrix.

It can be revisited by HTMP with the following components:

$$\begin{aligned}
\mathcal{A} &= [\mathbf{I}, \tilde{\mathbf{A}}, \mathbf{A}_f], \quad \mathcal{B} = [\mathbf{I}, \tilde{\mathbf{B}}^d, \mathbf{B}_f^d], \\
\tilde{\mathbf{Z}}_0^l &= (\mathbf{I} \odot \mathbf{I})\mathbf{Z}^{l-1}\mathbf{W}^l = \mathbf{Z}^{l-1}\mathbf{W}^l, \\
\tilde{\mathbf{Z}}_1^l &= (\tilde{\mathbf{A}} \odot \tilde{\mathbf{B}}^d)\mathbf{Z}^{l-1}\mathbf{W}^l = \hat{\tilde{\mathbf{A}}}\mathbf{Z}^{l-1}\mathbf{W}^l, \\
\tilde{\mathbf{Z}}_2^l &= (\mathbf{A}_f \odot \mathbf{B}_f^d)\mathbf{Z}^{l-1}\mathbf{W}^l = \hat{\mathbf{A}}_f\mathbf{Z}^{l-1}\mathbf{W}^l,
\end{aligned} \tag{31}$$

where the COMBINE function is adaptive weighted addition. Specifically, SimP-GCN aggregates messages from ego, raw and feature-similarity-based neighborhoods, and combines them with node-specific learnable weights. The feature-similarity-based neighborhoods can provide more homophilous messages to enhance the discriminability of the compatibility matrix. However, it's still limited by the amount of information on node features.

### B.2.8 FAGCN

FAGCN (Bo et al., 2021) proposes considering both low-frequency and high-frequency information simultaneously, and transferring them into the negative-allowable weights during message passing:

$$\mathbf{Z}_i^l = \mu \mathbf{Z}_i^0 + \sum_{j \in \mathcal{N}_i} \frac{\alpha_{ij}^G}{\sqrt{d_i d_j}} \mathbf{Z}_j^{l-1}, \tag{32}$$

where $\alpha_{ij}^G$ can be negative as follows:

$$\alpha_{ij}^G = \tanh(\mathbf{g}^T[\mathbf{X}_i \| \mathbf{X}_j]), \tag{33}$$

which can form a weight matrix:

$$\mathbf{P}_{ij}^{FAG} = \begin{cases} \alpha_{ij}^G, & \mathbf{A}_{ij} = 1, \\ 0, & \mathbf{A}_{ij} = 0. \end{cases} \tag{34}$$

It can be revisited by HTMP with the following components:

$$\begin{aligned}
\mathcal{A} &= [\mathbf{I}, \mathbf{A}], \quad \mathcal{B} = [\mathbf{I}, \mathbf{D}^{-\frac{1}{2}}\mathbf{P}^{FAG}\mathbf{D}^{-\frac{1}{2}}], \quad \mathbf{W}_0^l = \mathbf{W}_1^l = \mathbf{I}, \\
\tilde{\mathbf{Z}}_0^l &= (\mathbf{I} \odot \mathbf{I})\mathbf{Z}^0\mathbf{I} = \mathbf{Z}^0, \\
\tilde{\mathbf{Z}}_1^l &= (\mathbf{A} \odot \mathbf{D}^{-\frac{1}{2}}\mathbf{P}^{FAG}\mathbf{D}^{-\frac{1}{2}})\mathbf{Z}^{l-1}\mathbf{I} = \mathbf{D}^{-\frac{1}{2}}\mathbf{P}^{FAG}\mathbf{D}^{-\frac{1}{2}}\mathbf{Z}^{l-1},
\end{aligned} \tag{35}$$

where the COMBINE function is weighted addition, same as the matrix form of Eq 32. Specifically, FAGCN aggregates messages from node ego and raw neighborhood with negative-allowable weights. It has a similar form to GAT but allows for ego/neighbor separation and negative weights, which means the model can capture the difference between center nodes and neighbors.

### B.2.9 GGCN

GGCN (Yan et al., 2022) explicitly distinguishes between homophilous and heterophilous neighbors based on node similarities, and assigns corresponding positive and negative weights:

$$\mathbf{Z}^l = \alpha^l \left( \beta_0^l \hat{\mathbf{Z}}^l + \beta_1^l (\mathbf{S}_{pos}^l \odot \tilde{\mathbf{A}}_{\mathcal{T}}^l)\hat{\mathbf{Z}}^l + \beta_2^l (\mathbf{S}_{neg}^l \odot \tilde{\mathbf{A}}_{\mathcal{T}}^l)\hat{\mathbf{Z}}^l \right), \tag{36}$$

where $\hat{\mathbf{Z}}^l = \mathbf{Z}^{l-1}\mathbf{W}^l + b^l$, $\tilde{\mathbf{A}}_{\mathcal{T}}^l = \tilde{\mathbf{A}} \odot \mathcal{T}^l$ is an adjacency matrix weighted by the structure property, $\beta_0^l$, $\beta_1^l$ and $\beta_2^l$ are learnable scalars. The neighbors are distinguished by the cosine similarity of node representations with a threshold of 0:

$$\begin{aligned}
\mathbf{S}_{ij}^l &= \begin{cases} \text{Cosine}(\mathbf{Z}_i, \mathbf{Z}_j), & i \neq j \ \& \ \mathbf{A}_{ij} = 1, \\ 0, & \text{otherwise}. \end{cases}, \\
\mathbf{S}_{pos, \, ij}^l &= \begin{cases} \mathbf{S}_{ij}^l, & \mathbf{S}_{ij}^l > 0, \\ 0, & \text{otherwise}. \end{cases}, \\
\mathbf{S}_{neg, \, ij}^l &= \begin{cases} \mathbf{S}_{ij}^l, & \mathbf{S}_{ij}^l < 0, \\ 0, & \text{otherwise}. \end{cases}.
\end{aligned} \tag{37}$$

It can be revisited by HTMP with the following components:

$$
\begin{aligned}
\mathcal{A} &= [\mathbf{I}, \mathbf{A}_p, \mathbf{A}_n], \quad \mathcal{B} = [\mathbf{I}, \mathbf{S}_{pos}^l \odot \mathcal{T}^l, \mathbf{S}_{neg}^l \odot (T)^l], \\
\widetilde{\mathbf{Z}}_0^l &= (\mathbf{I} \odot \mathbf{I})\mathbf{Z}^{l-1}\mathbf{W}^l = \mathbf{Z}^{l-1}\mathbf{W}^l, \\
\widetilde{\mathbf{Z}}_1^l &= (\mathbf{A}_p \odot \mathbf{S}_{pos}^l \odot \mathcal{T}^l)\mathbf{Z}^{l-1}\mathbf{W}^l = (\mathbf{S}_{pos}^l \odot \mathcal{T}^l)\mathbf{Z}^{l-1}\mathbf{W}^l, \\
\widetilde{\mathbf{Z}}_2^l &= (\mathbf{A}_n \odot \mathbf{S}_{neg}^l \odot \mathcal{T}^l)\mathbf{Z}^{l-1}\mathbf{W}^l = (\mathbf{S}_{neg}^l \odot \mathcal{T}^l)\mathbf{Z}^{l-1}\mathbf{W}^l,
\end{aligned}
\tag{38}
$$

where $\mathbf{A}_p$ and $\mathbf{A}_n$ are discriminated by the representation similarities:

$$
\begin{aligned}
\mathbf{A}_{p,ij} &= \begin{cases} 1, & \mathbf{S}_{pos,ij}^l > 0 \,\&\, \mathbf{A}_{ij} = 1, \\ 0, & \text{otherwise}. \end{cases}, \\
\mathbf{A}_{n,ij} &= \begin{cases} 1, & \mathbf{S}_{neg,ij}^l < 0 \,\&\, \mathbf{A}_{ij} = 1, \\ 0, & \text{otherwise}. \end{cases}.
\end{aligned}
\tag{39}
$$

The COMBINE function is an adaptive weighted addition. Specifically, GGCN divides the raw neighborhood into positive and negative ones based on the similarities among node presentations. On this basis, it aggregates messages from node ego, positive and negative neighborhoods, and combines them with node-specific learnable weights. This approach allows for targeted processing for homophilous and heterophilous neighbors, yet can suffer from the accuracy of discrimination, which may lead to the accumulation of errors.

### B.2.10 ACM-GCN

ACM-GCN (Luan et al., 2022) introduces 3 channels (identity, low pass, and high pass) to capture different information and mixes them with node-wise adaptive weights:

$$
\mathbf{Z}^l = \text{diag}(\alpha_I^l)\mathbf{Z}^{l-1}\mathbf{W}_I^l + \text{diag}(\alpha_L^l)\hat{\mathbf{A}}\mathbf{Z}^{l-1}\mathbf{W}_L^l + \text{diag}(\alpha_H^l)(\mathbf{I} - \hat{\mathbf{A}})\mathbf{Z}^{l-1}\mathbf{W}_H^l,
\tag{40}
$$

where $\text{diag}(\alpha_I^l), \text{diag}(\alpha_L^l), \text{diag}(\alpha_H^l) \in \mathbb{R}^{N \times 1}$ are learnable weight vectors.

It can be revisited by HTMP with the following components:

$$
\begin{aligned}
\mathcal{A} &= [\mathbf{I}, \mathbf{A}, \mathbf{A}], \quad \mathcal{B} = [\mathbf{I}, \mathbf{B}^d, \mathbf{I} - \mathbf{B}^d], \\
\widetilde{\mathbf{Z}}_0^l &= (\mathbf{I} \odot \mathbf{I})\mathbf{Z}^{l-1}\mathbf{W}_I^l = \mathbf{Z}^{l-1}\mathbf{W}_I^l, \\
\widetilde{\mathbf{Z}}_1^l &= (\mathbf{A} \odot \mathbf{B}^d)\mathbf{Z}^{l-1}\mathbf{W}_L^l = \hat{\mathbf{A}}\mathbf{Z}^{l-1}\mathbf{W}_L^l, \\
\widetilde{\mathbf{Z}}_2^l &= (\mathbf{A} \odot (\mathbf{I} - \mathbf{B}^d))\mathbf{Z}^{l-1}\mathbf{W}_H^l = (\mathbf{I} - \hat{\mathbf{A}})\mathbf{Z}^{l-1}\mathbf{W}_H^l,
\end{aligned}
\tag{41}
$$

where the COMBINE function is adaptive weighted addition. Specifically, ACM-GCN aggregates node ego, low-frequency, and high-frequency messages from ego and raw neighborhoods, and combines them with node-wise adaptive weights. With simple but effective designs, ACM-GCN achieves outstanding performance, which shows that complicated designs are not necessary.

### B.2.11 GPR-GNN

GPR-GNN (Chien et al., 2021) integrates messages from multiple-order neighborhoods with adaptive weights:

$$
\mathbf{Z} = \sum_{l=0}^{K} \gamma_l \mathbf{Z}^l, \quad \mathbf{Z}^l = \tilde{\mathbf{D}}^{-\frac{1}{2}}\tilde{\mathbf{A}}\tilde{\mathbf{D}}^{-\frac{1}{2}}\mathbf{Z}^{l-1},
\tag{42}
$$

where $\gamma_l$ are learnable weights.

It can be revisited by HTMP with the following components:

$$
\begin{aligned}
\mathbf{A}_0 &= \mathbf{A}, \quad \mathbf{B}_0 = \mathbf{B}^d, \quad \mathbf{W}_0^l = \mathbf{I}, \\
\mathbf{Z}^l &= \mathbf{Z}_0^l = (\mathbf{A}_0 \odot \mathbf{B}_0)\mathbf{Z}^{l-1}\mathbf{W}_0^l = \hat{\mathbf{A}}\mathbf{Z}^{l-1}, \\
\mathbf{Z} &= \text{Fuse}(\mathbf{Z}^l) = \sum_{l=0}^{K} \gamma_l \mathbf{Z}^l.
\end{aligned}
\tag{43}
$$

where the Fuse function is adaptive weighted addition.

Specifically, GPR-GNN has a Fuse function with adaptive weights and no feature transformation between layers, while other settings are the same as GCN. It can gather messages from neighbors of different hops and construct more discriminative representations.

### B.2.12 ORDEREDGNN

OrderedGNN (Song et al., 2023) is a SOTA method that introduces a node-wise adaptive dimension concatenation function to combine messages from neighbors of different hops:

$$\mathbf{Z}^l = \mathbf{P}_d^l \odot \mathbf{Z}^{l-1} + (1 - \mathbf{P}_d^l) \odot (\hat{\mathbf{A}}\mathbf{Z}^{l-1}), \tag{44}$$

where $\mathbf{P}_d \in \mathbb{R}^{N \times d_r}$ is designed to be matrix with each line $\mathbf{P}_{d,i}^l$ being a dimension indicate vector, which starts with continuous 1s while the others be 0s. In practice, to keep the differentiability, it's "soften" as follows:

$$\hat{\mathbf{P}}_d^l = \text{cumsum}_{\leftarrow} \left( \text{softmax} \left( f_\xi^l \left( \mathbf{Z}^{l-1}, \hat{\mathbf{A}}\mathbf{Z}^{l-1} \right) \right) \right),$$
$$\mathbf{P}_d^l = \text{SOFTOR}(\mathbf{P}_d^{l-1}, \hat{\mathbf{P}}_d^l), \tag{45}$$

where $f_\xi^l$ is a learnable layer that fuses two messages.

It can be revisited by HTMP with the following components:

$$\mathcal{A} = [\mathbf{I}, \mathbf{A}], \quad \mathcal{B} = [\mathbf{I}, \mathcal{B}^d], \quad \mathbf{W}_0^l = \mathbf{W}_1^l = \mathbf{I},$$
$$\widetilde{\mathbf{Z}}_0^l = (\mathbf{I} \odot \mathbf{I})\mathbf{Z}^{l-1} = \mathbf{Z}^{l-1},$$
$$\widetilde{\mathbf{Z}}_1^l = (\mathbf{A} \odot \mathbf{B}^d)\mathbf{Z}^{l-1} = \hat{\mathbf{A}}\mathbf{Z}^{l-1}, \tag{46}$$

where the COMBINE function is concatenated with node-wise adaptive dimensions. Specifically, in each layer, OrderedGNN aggregates messages from node ego and raw neighborhood and concatenates them with learnable dimensions. Combined with the multi-layer architecture, this approach can aggregate messages from neighbors of different hops and combine them not only with adaptive contributions but also as separately as possible.

### B.3 ANALYSIS AND ADVICE FOR DESIGNING MODELS

The HTMP mechanism splits the message-passing mechanism of HTGNNs into multiple modules, establishing connections among methods. For instance, most messages passing in HTGNNs have personalized processing for nodes. Some methods (Du et al., 2022; Bo et al., 2021; Jin et al., 2021a; Suresh et al., 2021) utilize the learnable aggregation guidance and some others (Jin et al., 2021b; Luan et al., 2022; Song et al., 2023; Yan et al., 2022) count on learnable COMBINE functions. Though neighborhood redefining is commonly used in HTGNNs, there are also many methods (Du et al., 2022; Bo et al., 2021; Luan et al., 2022; Chien et al., 2021; Song et al., 2023) using only raw neighborhoods to handle heterophily and achieve good performance. Degree averaging, which plays the role of a low-pass filter to capture the smooth signals, can still work well in many HTGNNs (Zhu et al., 2020; Jin et al., 2021b; Pei et al., 2020; Abu-El-Haija et al., 2019; Chien et al., 2021). High-order neighbor information may be helpful in heterophilous graphs. Existing HTGNNs utilize it in two ways: directly defining high-order (Zhu et al., 2020; Jin et al., 2021a; Abu-El-Haija et al., 2019; Wang et al., 2022) or even full-connected (Li et al., 2022) neighborhood indicators and by the multi-layer architecture of message passing (Chien et al., 2021; Song et al., 2023).

With the aid of HTMP, we can revisit existing methods from a unified and comprehensible perspective. An obvious observation is that *the coordination among designs is important while good combinations with easy designs can also achieve wonderful results.* For instance, in ACM-GCN (Luan et al., 2022), the separation and adaptive addition of ego, low-frequency, and high-frequency messages can accommodate the personalized conditions of each node. OrderedGNN's design (Song et al., 2023), which includes an adaptive connection mechanism, ego/neighbor separation, and multi-layer architecture, allows discrete and adaptive combinations of messages from multi-hop neighborhoods. This advises us to *take into account all components simultaneously* when designing models. As an illustration, please be cautious about using multiple learnable components. Also, here are some

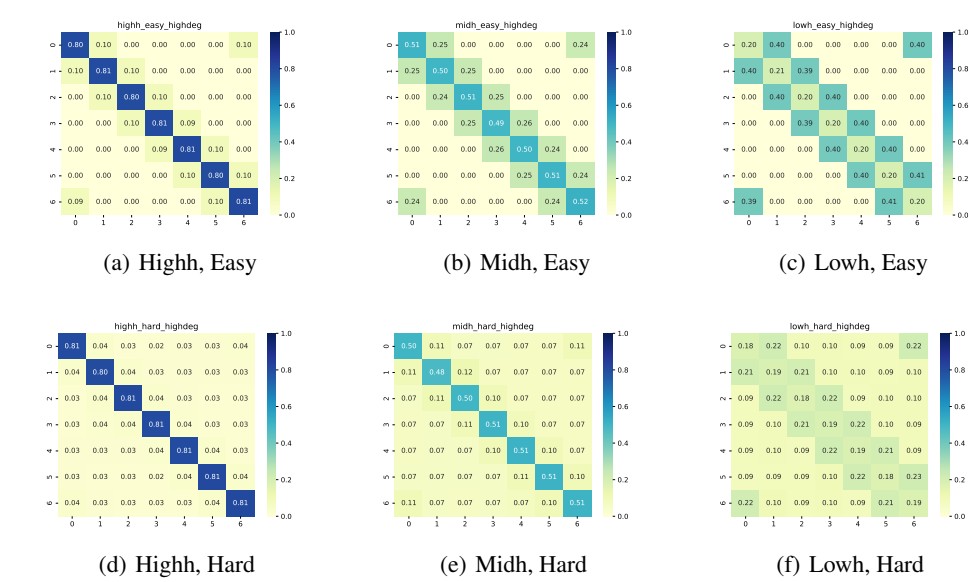

(a) Highh, Easy     (b) Midh, Easy     (c) Lowh, Easy

(d) Highh, Hard     (e) Midh, Hard     (f) Lowh, Hard

Figure 5: The visualization of compatibility matrix on synthetic graphs.

additional model design tips and considerations. Please *separate the messages from node ego and neighbors*. When combining them afterward, whether by weighted addition or concatenation, this approach is at least harmless if not beneficial, especially when dealing with heterophilous graphs. Last but not least, try to design a model capable of *personalized handling different nodes*. Available components include but are not limited to, custom-defined neighborhood indicators, aggregation guidance with adaptive weights or estimated relationships, and learnable COMBINE functions. This is to accommodate the diversity and sparsity of neighborhoods that nodes in real-world graphs may have.

## C    THE DETAIL OF EXPERIMENTS ON SYNTHETIC DATASETS

To explore the performance impact of homophily level, node degrees, and compatibility matrix (CMs) on simple GNNs, we conduct some experiments on synthetic datasets.

### C.1    SYNTHETIC DATASETS

We construct synthetic graphs considering the factors of homophily, CMs, and degrees. For homophily, we set 3 levels including Lowh (0.2), Midh (0.5), and Highh (0.8). For CMs, we set two levels of discriminability, including Easy and Hard. For degrees, we set two levels including Lowdeg (4) and Highdeg (18). Note that with a certain homophily level, we can only control the non-diagonal elements of CMs. Thus, there are a total of 12 synthetic graphs following the above settings. These synthetic graphs are based on the Cora dataset, which provides node features and labels, which means, only the edges are constructed. We visualize the CMs of these graphs in Figure 5. Since there is no significant difference in CMs between low-degree and high-degree, we only plot the high-degree ones. Further, the edges are randomly constructed under the guidance of these CMs and degrees to form the synthetic graphs.

### C.2    EXPERIMENTS ON SYNTHETIC DATASETS

We use GCN to analyze the performance impact of the above factors. The semi-supervised node classification performance of GCN is shown in Table 5 while the baseline performance of MLP (72.54 ± 2.18) is the same among these datasets since their difference is only on edges. From these results, we have some observations: (1) high homophily is not necessary, GCN can also work well on low homophily but discriminative CM; (2) low degrees have a negative impact on performance, especially

Table 5: Node classification accuracy of GCN on synthetic datasets.

| Factors | Highh, Esay | Highh, Hard | Midh, Easy | Midh, Hard | Lowh, easy | Lowh, Hard |
|---------|-------------|-------------|------------|------------|------------|------------|
| **Highd** | $99.15 \pm 0.35$ | $99.48 \pm 0.24$ | $86.42 \pm 4.13$ | $90.52 \pm 1.05$ | $89.34 \pm 2.19$ | $39.22 \pm 2.34$ |
| **Lowd** | $89.98 \pm 1.59$ | $91.25 \pm 0.85$ | $70.85 \pm 1.59$ | $70.20 \pm 1.41$ | $56.46 \pm 2.63$ | $40.91 \pm 1.75$ |

when the CMs are relatively weak discriminative, this also indicates that nodes with lower degrees are more likely to have confused neighborhoods; and (3) when dealing with nodes with confused neighborhoods, GCN may contaminate central nodes with their neighborhoods' messages, which leads to performance worse than MLP. This once again reminds us of the importance of ego/neighbor separation.

## D  THEORETICAL PROOF

To prove Theorem 1, we start with an assumption:

**Assumption 1.** *The semantic neighborhood $C^{nb}$ of each node follows a class-specific distribution guided by CM, where $C_i^{nb} = \frac{1}{d_i} \sum_{j \in \mathcal{N}(i)} C_j$ indicates the proportion of neighbors from each class in node $i$'s neighborhood.*

According to Assumption 1, the discriminability in CM is positively correlated with the discriminability in semantic neighborhoods. Thus, if the message-passing mechanism is able to preserve the discriminability of the semantic neighborhood in the obtained representations, then Theorem 1 holds. It would be sufficient if each distinct semantic neighborhood corresponds to a different output representation, in other words, the message-passing mechanism is an injective function for modeling semantic neighborhoods. We further state two assumptions:

**Assumption 2.** *The input node messages $\mathbf{Z}^{l-1}$ of the message-passing layer exhibit clustering characteristics, where the average distance within a class is significantly smaller than the average distance between different classes.*

This implies that the input messages of nodes from the same class are linearly correlated within a certain range of error.

**Assumption 3.** *The clustering centers of each class's input messages exist, formatted as $K$ prototypes $\{c_k | k \in 1, ..., K\}$.*

Taking the most general mean aggregation as an example, we have the following theorem:

**Theorem 2.** *Let $MEAN(\{\mathbf{Z}_j^{l-1} | j \in \mathcal{N}(i)\})$ be mean operator that aggregate neighbor messages for node $i$, $c_k$ be the prototype of $k$-th class. Function $MEAN(\{\mathbf{Z}_j^{l-1} | j \in \mathcal{N}(i)\})$ is approximately injective if it is satisfied that all class prototypes $c_k$ are orthogonal to each other.*

The injectivity ensures that each element in the domain of the input (i.e. semantic neighborhoods and neighbor messages) has a distinct and unique output in the output domain. We find that as long as the conditions of Theorem 2 are satisfied, the mean aggregation can be regarded as an injective function within a certain range of error. Thus, the whole message-passing mechanism can be an approximately injective function for modeling the semantic neighborhoods when the COMBINE function is also injective, which can be easily satisfied. In practice, the orthogonality of prototypes is hard to be satisfied completely but the difference among prototypes is still significant. Thus, even if the message-passing mechanism is not completely injective, most of the discriminability can be preserved, making Theorem 1 hold.

*Proof of Theorem 2.* Firstly, we have the following lemma:

**Lemma 1.** *Injectivity is equivalent to null space equals $\{0\}$. Let $T \in \mathcal{L}(V, W)$, $T$ is injective if and only if $null(T) = \{0\}$.*

*Proof of Lemma 1.* We make the proof of Lemma 1 from the perspectives of both its sufficiency and necessity as follows:

**Sufficiency**: First, suppose $T$ is injective. We want to prove that $null(T) = \{0\}$. We already know that $\{0\} \subset null(T)$. Suppose $v \in null(T)$, then $T(v) = 0 = T(0)$. Because $T$ is injective, the equation implies that $v = 0$. Thus we can conclude that $null(T) = \{0\}$, as desired.

**Necessity**: Suppose $null(T) = \{0\}$, $u, v \in V$. If $T(u) = T(v)$, then $T(u) - T(v) = T(u - v) = 0$. Thus $u - v = 0$, which implies that $u = v$. Hence $T$ is injective, as desired. $\qquad\square$

Having Lemma 1 proved, now we express the mean aggregation in the form of $PZ^{nb} = b$, where $P \in \mathbb{R}^{1 \times |\mathcal{N}(i)|}$ denotes the mean aggregation operator, $Z^{nb} \in \mathbb{R}^{|\mathcal{N}(i)| \times d_r}$ is the matrix consist of neighbor messages and $b$ is the resulting representation. Assuming that the messages of neighbors from the same class are linearly dependent, we can rewrite the equation as $P'Z^p \approx b$, where $P' \in \mathbb{R}^{1 \times K}$ is a weighted mean operator, $Z^p \in \mathbb{R}^{K \times d_r}$ is a matrix consisting of the prototypes $\{c_k | k \in 1, ..., K\}$ of $K$ classes.

The injectivity of mean aggregation operator $P$ involves considering the solution for $P'Z^p = 0$. Clearly, if it is satisfied that all $c_k$ are orthogonal to each other, the null space $null(P') = \{0\}$, indicating that the mean aggregation operator is approximately injective, as desired. $\qquad\square$

The above analyses provide theoretical support for Observation 1 and Observation 2. Based on Theorem 1, VMP can work well with discriminative CM regardless of homophily levels and HTMP can achieve better performance by enhancing the discriminability of CM.

# E EMPIRICAL EVIDENCE FOR THE CONJECTURE ABOUT CM

In this part, we give the details of the empirical evidence for the conjecture mentioned in Sec 4. The detailed method of ACM-GCN and GPR-GNN can be seen in B.2.10 and B.2.11. The desired CMs are obtained as follows:

- For ACM-GCN, we leverage the learned weights in the COMBINE function to rebuild a weighted adjacency matrix $\mathbf{A}^{acm}$ based on the low-pass filter $\hat{\mathbf{A}}$ and high-pass filter $\mathbf{I} - \hat{\mathbf{A}}$, then regard $\mathbf{A}^{acm}$ as the neighborhood and calculate the desired CM.
- For GPR-GNN, we utilize the leaned weights in the FUSE function to rebuild a weighted adjacency matrix $\mathbf{A}^{gpr}$ based on the multi-hop adjacency matrixes $[\mathbf{I}, \mathbf{A}, \mathbf{A}^2, ..., \mathbf{A}^k]$ then regard $\mathbf{A}^{gpr}$ as the neighborhood and calculate the desired CM.

# F ADDITIONAL DETAILED IMPLEMENTATION OF CMGNN

**Overall Message Passing Mechanism.** The overall message passing mechanism in CMGNN is formatted as follows:

$$\mathbf{Z}^l = \text{diag}(\alpha_0^l)\mathbf{Z}^{l-1}\mathbf{W}_0^l + \text{diag}(\alpha_1^l)\hat{\mathbf{A}}\mathbf{Z}^{l-1}\mathbf{W}_1^l + \text{diag}(\alpha_2^l)(\mathbf{A}^{sup} \odot \mathbf{B}^{sup})\mathbf{Z}^{l-1}\mathbf{W}_2^l,$$

$$\mathbf{Z} = \overset{L}{\underset{l=0}{\|}} \mathbf{Z}^l, \tag{47}$$

where $\text{diag}(\alpha_0^l), \text{diag}(\alpha_1^l), \text{diag}(\alpha_2^l) \in \mathbb{R}^{N \times 1}$ are the learned combination weights introduced below.

**COMBNIE Function with Adaptive Weights.** Firstly, we list the aggregated messages $\widetilde{\mathbf{Z}}_r^l$ from 3 neighborhoods:

$$\widetilde{\mathbf{Z}}_0^l = \mathbf{Z}^{l-1}\mathbf{W}_0^l, \ \widetilde{\mathbf{Z}}_1^l = \hat{\mathbf{A}}\mathbf{Z}^{l-1}\mathbf{W}_1^l,$$

$$\widetilde{\mathbf{Z}}_2^l = (\mathbf{A}^{sup} \odot \mathbf{B}^{sup})\mathbf{Z}^{l-1}\mathbf{W}_2^l. \tag{48}$$

The combination weights are learned by an MLP with Softmax:

$$[\alpha_0^l, \alpha_1^l, \alpha_2^l] = \text{Softmax}(\text{Sigmoid}([\mathbf{Z}_0^l \| \mathbf{Z}_1^l \| \mathbf{Z}_2^l \| \mathbf{d}]\mathbf{W}_{att}^l)\mathbf{W}_{mix}^l), \tag{49}$$

---

**Algorithm 1** Algorithm of CMGNN

---

**Require:** Graph $\mathcal{G} = (\mathcal{V}, \mathcal{E}, \mathbf{X}, \mathbf{A}, \mathbf{Y})$, loss weight $\lambda$, epoch $E$

**Ensure:** Predicted labels $\hat{\mathbf{Y}}$

1: Initialize the soft predicted labels $\tilde{C}$ with other elements $\frac{1}{K}$.
2: Construct class prototypes as additional virtual neighbors for all nodes via Eq.4.
3: **for** iteration 1, 2, ..., $E$ **do**
4:     Obtain the input representations for the first layer via Eq.13.
5:     Estimate the compatibility matrix via Eq.5, Eq.6, Eq.7, and Eq.8.
6:     Obtain the output representations through the CM-aware message-passing mechanism via
    Eq.12, or the detailed version Eq.47, Eq.48, Eq.49, and Eq.50.
7:     Obtain the predicted logits (soft label) $\tilde{C}$ via Eq.14.
8:     Calculate loss $\mathcal{L}$ via Eq.15.
9:     Back-propagation $\mathcal{L}$ to optimize the weights of networks.
10:     **if** the performance in the validation set improved **then**
11:       update the compatibility matrix with current soft predicted label $\tilde{\mathbf{C}}$.
12:     **end if**
13: **end for**
14: Obtain the predicted labels $\hat{\mathbf{Y}}$ via $\hat{\mathbf{Y}} = \text{Softmax}(\tilde{\mathbf{C}})$.
15: **return** $\hat{\mathbf{Y}}$

---

where $\mathbf{W}_{att}^l \in \mathbb{R}^{(3d_r+1)\times 3}$ and $\mathbf{W}_{mix}^l \in \mathbb{R}^{3\times 3}$ are two learnable weight matrixes, $\mathbf{d}$ is the node degrees which may be helpful to weights learning.

**The Message Passing of Supplementary Prototypes.** Specifically, the virtual prototype nodes are viewed as additional nodes, which have the same message-passing mechanism as real nodes:

$$\mathbf{Z}^{ptt,l} = \text{diag}(\alpha_0^{ptt,l})\mathbf{Z}^{ptt,l-1}\mathbf{W}_0^l + \text{diag}(\alpha_1^{ptt,l})\hat{\mathbf{A}}^{ptt}\mathbf{Z}^{ptt,l-1}\mathbf{W}_1^l$$
$$+ \text{diag}(\alpha_2^{ptt,l})(\mathbf{A}^{ptt,sup} \odot \mathbf{B}^{ptt,sup})\mathbf{Z}^{ptt,l-1}\mathbf{W}_2^l, \quad (50)$$
$$\mathbf{Z}^{ptt} = \overset{L}{\underset{l=0}{\|}} \mathbf{Z}^{ptt,l},$$

where $\mathbf{A}^{sup,ptt} = \mathbf{1} \in \mathbb{R}^{K\times K}$ and $\mathbf{B}^{sup,ptt} = \hat{\mathbf{C}}^{ptt}\hat{\mathbf{M}}$ are similar with those of real nodes.

## G MORE DETAIL ABOUT THE BENCHMARK

In this section, we describe the details of the new benchmarks, including (i) the reason why we need a new benchmark: drawbacks of existing datasets; (ii) detailed descriptions of newly organized datasets; (iii) baseline methods and the codebase; and (iv) details of obtaining benchmark performance.

### G.1 DRAWBACKS IN EXISTING DATASETS

As mentioned in Platonov et al. (2023), the widely used datasets Cornell, Texas, and Wisconsin[2] have a too small scale for evaluation. Further, the original datasets Chameleon and Squirrel have an issue of data leakage, where some nodes may occur simultaneously in both training and testing sets. Then, the splitting ratio of training, validation, and testing sets are different across various datasets, which is ignored in previous works.

Therefore, to build a comprehensive and fair benchmark for model effectiveness evaluation, we will newly organize 10 datasets with unified splitting across various homophily values in the next Subsection G.2.

### G.2 NEWLY ORGANIZED DATASETS

In our benchmark, we adopt ten different types of publicly available datasets with a unified splitting setting (48%/32%/20% for training/validation/testing) for fair model comparison, including **Roman-**

---

[2]https://www.cs.cmu.edu/afs/cs.cmu.edu/project/theo-11/www/wwkb

**Empire** (Platonov et al., 2023), **Amazon-Ratings** (Platonov et al., 2023), **Chameleon-F** (Platonov et al., 2023), **Squirrel-F** (Platonov et al., 2023), **Actor** (Pei et al., 2020), **Flickr** (Liu et al., 2021), **BlogCatalog** (Liu et al., 2021), **Wikics** (Mernyei & Cangea, 2020), **Pubmed** (Sen et al., 2008), and **Photo** (Shchur et al., 2018). The datasets have a variety of homophily values from low to high. The statistics and splitting of these datasets are shown in Table 6. The detailed description of the datasets is as follows:

Table 6: Statistics and splitting of the experimental benchmark datasets.

| Dataset | Nodes | Edges | Attributes | Classes | Avg. Degree | Undirected | Homophily | Train / Valid / Test |
|---|---|---|---|---|---|---|---|---|
| Roman-Empire | 22,662 | 65,854 | 300 | 18 | 2.9 | ✓ | 0.05 | 10,877 / 7,251 / 4,534 |
| Amazon-Ratings | 24,492 | 186,100 | 300 | 5 | 7.6 | ✓ | 0.38 | 11,756 / 7,837 / 4,899 |
| Chameleon-F | 890 | 13,584 | 2,325 | 5 | 15.3 | ✗ | 0.25 | 427 / 284 / 179 |
| Squirrel-F | 2,223 | 65,718 | 2,089 | 5 | 29.6 | ✗ | 0.22 | 1,067 / 711 / 445 |
| Actor | 7,600 | 30,019 | 932 | 5 | 3.9 | ✗ | 0.22 | 3,648 / 2,432 / 1,520 |
| Flickr | 7,575 | 479,476 | 12,047 | 9 | 63.3 | ✓ | 0.24 | 3,636 / 2,424 / 1,515 |
| BlogCatalog | 5,196 | 343,486 | 8,189 | 6 | 66.1 | ✓ | 0.40 | 2,494 / 1,662 / 1,040 |
| Wikics | 11,701 | 431,206 | 300 | 10 | 36.9 | ✓ | 0.65 | 5,616 / 3,744 / 2,341 |
| Pubmed | 19,717 | 88,651 | 500 | 3 | 4.5 | ✓ | 0.80 | 9,463 / 6,310 / 3,944 |
| Photo | 7,650 | 238,162 | 745 | 8 | 31.1 | ✓ | 0.83 | 3,672 / 2,448 / 1,530 |

- **Roman-Empire**[3] (Platonov et al., 2023) is derived from the extensive article on the Roman Empire found on the English Wikipedia, chosen for its status as one of the most comprehensive entries on the platform. It contains 22,662 nodes and 65,854 edges between nodes. Each node represents an individual word from the text, with the total number of nodes mirroring the length of the article. An edge between two nodes is established under one of two conditions: the words are sequential in the text or they are linked in the sentence's dependency tree, indicating a grammatical relationship where one word is syntactically dependent on the other. Consequently, the graph is structured as a chain graph, enriched with additional edges that represent these syntactic dependencies. The graph encompasses a total of 18 distinct node classes, with each node being equipped with 300-dimensional attributes obtained by fastText word embeddings (Grave et al., 2018).

- **Amazon-Ratings**[3] (Platonov et al., 2023) is sourced from the Amazon product co-purchasing network metadata dataset (Jure, 2014). It contains 24,492 nodes and 186,100 edges between nodes. The nodes within this graph represent products, encompassing a variety of categories such as books, music CDs, DVDs, and VHS video tapes. An edge between nodes signifies that the respective products are often purchased together. The objection is to forecast the average rating assigned to a product by reviewers, with the ratings being categorized into five distinct classes. For the purpose of node feature representation, we have utilized the 300-dimensional mean values derived from fastText word embeddings (Grave et al., 2018), extracted from the textual descriptions of the products.

- **Chameleon-F** and **Squirrel-F**[3] (Platonov et al., 2023) are specialized collections of Wikipedia page-to-page networks (Rozemberczki et al., 2021), of which the data leakage nodes are filtered out by Platonov et al. (2023). Within these datasets, each node symbolizes a web page, and edges denote the mutual hyperlinks that connect them. The node features are derived from a selection of informative nouns extracted directly from Wikipedia articles. For the purpose of classification, nodes are categorized into five distinct groups based on the average monthly web traffic they receive. Specifically, Chameleon-F contains 890 nodes and 13,584 edges between nodes, with each node being equipped with 2,325-dimensional features. Squirrel-F contains 2,223 nodes and 65,718 edges between nodes, with each node being equipped with a 2,089-dimensional feature vector.

- **Actor**[4] (Pei et al., 2020) is an actor-centric induced subgraph derived from the broader film-director-actor-writer network, as originally presented by Tang et al. (2009). In this refined network, each node corresponds to an individual actor, and the edges signify the co-occurrence of these actors on the same Wikipedia page. The node features are identified through the presence of certain keywords found within the actors' Wikipedia entries. For the purpose of classification, the actors are organized into five distinct categories based on the words of the actor's Wikipedia. Statistically,

---

[3]https://github.com/yandex-research/heterophilous-graphs/tree/main/data
[4]https://github.com/bingzhewei/geom-gcn/tree/master/new_data/film

it contains 7,600 nodes and 30,019 edges between nodes, with each node being equipped with a 932-dimensional feature vector.

- **Flickr** and **Blogcatalog**[5] (Liu et al., 2021) are two datasets of social networks, originating from the blog-sharing platform BlogCatalog and the photo-sharing platform Flickr, respectively. Within these datasets, nodes symbolize the individual users of the platforms, while links signify the followship relations that exist between them. In the context of social networks, users frequently create personalized content, such as publishing blog posts or uploading and sharing photos with accompanying tag descriptions. These textual contents are consequently treated as attributes associated with each node. The classification objection is to predict the interest group of each user. Specifically, Flickr contains 7,575 nodes and 479,476 edges between nodes. The graph encompasses a total of 9 distinct node classes, with each node being equipped with a 12047-dimensional attribute vector. BlogCatalog contains 5,196 nodes and 343,486 edges between nodes. The graph encompasses a total of 6 distinct node classes, with each node being equipped with 8189-dimensional attributes.

- **Wikics**[6] (Mernyei & Cangea, 2020) is a dataset curated from Wikipedia, specifically designed for benchmarking the performance of GNNs. It is meticulously constructed around 10 distinct categories that represent various branches of computer science, showcasing a high degree of connectivity. The node features are extracted from the text of the associated Wikipedia articles, leveraging the power of pretrained GloVe word embeddings (Pennington et al., 2014). These features are computed as the average of the word embeddings, yielding a comprehensive 300-dimensional representation for each node. The dataset encompasses a substantial network of 11,701 nodes interconnected by 431,206 edges.

- **Pubmed**[7] (Sen et al., 2008) is a classical citation network consisting of 19,717 scientific publications with 44,338 links between them. The text contents of each publication are treated as their node attributes, and thus each node is assigned a 500-dimensional attribute vector. The target is to predict which of the paper categories each node belongs to, with a total of 3 candidate classes.

- **Photo**[8] (Shchur et al., 2018) is one of the Amazon subset network from Shchur et al. (2018). Nodes in the graph represent goods and edges represent that two goods are frequently bought together. Given product reviews as bag-of-words node features, each node is assigned a 745-dimensional feature vector. The task is to map goods to their respective product category. It contains 7,650 nodes and 238,162 edges between nodes. The graph encompasses a total of 8 distinct product categories.

### G.3 BASELINE METHODS AND THE CODEBASE

For comprehensive comparisons, we choose 17 representative baseline methods as in the benchmark, which can be categorized into four main groups of works as follows:

(i) **Shallow Model**: MLP;

(ii) **Homopihlous Graph Neural Networks**: GCN (Kipf & Welling, 2017), GAT (Veličković et al., 2018), APPNP (Gasteiger et al., 2019), and GCNII (Chen et al., 2020);

(iii) **Heterophilous Graph Neural Networks**: H2GCN (Zhu et al., 2020), MixHop (Abu-El-Haija et al., 2019), GBK-GNN (Du et al., 2022), GGCN (Yan et al., 2022), GloGNN (Li et al., 2022), HOGGCN (Wang et al., 2022), GPR-GNN (Chien et al., 2021), ACM-GCN (Luan et al., 2022), and OrderedGNN (Song et al., 2023);

(iv) **Compatibility Matrix-based Models**: CLP (Zhong et al., 2022), EPFGNN (Wang et al., 2021), and CPGNN (Zhu et al., 2021a).

Detailed descriptions of some of these methods can be found in Appendix B.2. To explore the performance of baseline methods on newly organized datasets and facilitate future expansions, we collect the official/reproduced codes from GitHub and integrate them into a unified codebase. Specifically, all methods share the same data loaders and evaluation metrics. One can easily run

---

[5] https://github.com/TrustAGI-Lab/CoLA/tree/main/raw_dataset
[6] https://github.com/pmernyei/wiki-cs-dataset
[7] https://linqs.soe.ucsc.edu/datac
[8] https://github.com/shchur/gnn-benchmark

different methods with only parameters changing within the codebase. The codebase is based on the widely used PyTorch[9] framework, supporting both DGL[10] and PyG[11]. Detailed usages of the codebase are available in the Readme file of the codebase.

### G.4 Details of Obtaining Benchmark Performance

Following the settings in existing methods, we construct 10 random splits (48%/32%/20% for train/valid/test) for each dataset and report the average performance among 10 runs on them along with the standard deviation.

For all baseline methods except MLP, GCN, and GAT, we conduct parameter searches within the search space recommended by the original papers. The searches are based on the NNI framework with an anneal strategy. We use Adam as the optimizer for all methods. Each method has dozens of search trails according to their time costs and the best performances are reported. The currently known optimal parameters of each method are listed in the codebase. We run these experiments on NVIDIA GeForce RTX 3090 GPU with 24G memory. The out-of-memory error during model training is reported as OOM in Table 2.

## H  More Details about Experiments

In this section, we describe the additional details of the experiments, including experimental settings and results.

### H.1 Additional Experimental Settings

Our method has the same experimental settings within the benchmark, including datasets, splits, evaluations, hardware, optimizer, and so on as in Appendix G.4.

**Parameters Search Space.** We list the search space of parameters in Table 7, where patience is for the maximum epoch early stopping, n_hidden is the embedding dimension of hidden layers as well as the representation dimension $d_r$, relu_varient decides ReLU applying before message aggregation or not as in Luan et al. (2022), structure_info determines whether to use structure information as supplement node features or not.

Table 7: Parameter search space of our method.

| Parameters | Range |
|---|---|
| learning rate | {0.001, 0.005, 0.01, 0.05} |
| weight_decay | {0, 1e-7, 5e-7, 1e-6, 5e-6, 5e-5, 5e-4} |
| patience | {200, 400} |
| dropout | [0, 0.9] |
| $\lambda$ | {0, 0.01, 0.1, 1, 10} |
| layers | {1, 2, 4, 8} |
| n_hidden | {32, 64, 128, 256} |
| relu_variant | {True, False} |
| structure_info | {True, False} |

**Ablation Study.** In the ablation study, there are three variants of our methods: without SM, without DL, without SM, and DL. For "without SM", we delete the supplementary messages during message passing, using only messages from node ego and raw neighborhood for combination. For "without DL", we simply set $\lambda = 0$ to delete the discrimination loss. For "without SM and DL", we just combine the above two settings.

---

[9] https://pytorch.org
[10] https://www.dgl.ai
[11] https://www.pyg.org

Table 8: Node classification accuracy (%) and computational cost (min) comparison on large-scale graphs. The error bar (±) denotes the standard deviation of results over 10 trial runs. The best and second-best results in each dataset are highlighted in **bold** font and underlined.

| Dataset | snap-patents | | pokec | | genius | |
|---|---|---|---|---|---|---|
| **Homo.** | 0.073 | | 0.445 | | 0.618 | |
| **Nodes** | 2,923,922 | | 1,632,803 | | 421,961 | |
| **Edges** | 13,975,788 | | 30,622,564 | | 984,979 | |
| **Classes** | 5 | | 2 | | 2 | |
| **Method** | **acc (%)** | **cost (min)** | **acc (%)** | **cost (min)** | **acc (%)** | **cost (min)** |
| MLP | $31.30 \pm 0.07$ | 37 | $62.29 \pm 0.09$ | 75 | $82.54 \pm 0.14$ | 2 |
| GCN | $37.97 \pm 0.04$ | 87 | $70.12 \pm 0.13$ | 96 | $84.25 \pm 0.13$ | 12 |
| GAT | $38.42 \pm 0.24$ | 80 | $73.76 \pm 0.38$ | 287 | $81.89 \pm 0.39$ | 16 |
| OrderedGNN | $38.43 \pm 0.22$ | 305 | $75.17 \pm 0.18$ | 273 | $85.15 \pm 0.65$ | 63 |
| GCNII | $40.90 \pm 0.18$ | 562 | $78.18 \pm 0.49$ | 522 | $84.99 \pm 0.48$ | 188 |
| **CMGNN** | **$62.86 \pm 0.38$** | 148 | **$81.74 \pm 0.50$** | 393 | **$85.44 \pm 0.20$** | 27 |

## H.2 ADDITIONAL EXPERIMENTAL RESULTS AND ANALYSES

In this subsection, we show some additional experimental results and analyses.

### H.2.1 DETAILED ANALYSIS ABOUT THE COMPARISON BETWEEN CMGNN AND EXISTING CM-BASED METHODS

Specifically, CMGNN achieves better performances and benefits from the approach of utilizing CM in the following aspects: (i) Better robustness for low-quality pseudo labels: Existing CM-based methods utilize CM to guide the weights of propagation, which can lead to error accumulation with inaccurate pseudo labels. This is a common limitation of CM-based methods. In CMGNN, the CM is used to construct supplementary messages while original neighborhoods are preserved, mitigating the impact of inaccurate pseudo labels. (ii) Unlock the effectiveness of CM for low-degree nodes: Existing CM-based methods redefine pair-wise relations only for existing edges, limiting the effectiveness of CMs for low-degree nodes. In CMGNN, virtual neighbors can provide prototype messages from every class, enhancing neighborhood messages for low-degree or even isolated nodes. (iii) More accurate estimation of CM: While existing CM-based methods take naive approaches to estimate or initialize CM, CMGNN considers the effects of node degrees and model prediction confidence, resulting in more accurate CM estimation, especially in real-world situations. Additionally, CM in CMGNN is continuously updated with more accurate pseudo labels, creating a positive cycle.

### H.2.2 EXPERIMENTS ON LARGE-SCALE GRAPHS

To further evaluate the scalability of CMGNN, we conduct additional experiments on 3 large-scale datasets including snap-patents, pokec and genius (Lim et al., 2021), comparing with representative baselines. The details of the datasets and results are listed in Table H.2.1. As a result, CMGNN can also achieve superior performance while maintaining good efficiency, especially on snap-patents with 22% improvements, which demonstrates great scalability.

### H.2.3 ABLATION STUDY ON ADDITIONAL STRUCTURAL FEATURES

Utilizing additional structural features is a common approach in heterophilous GNNs that offers another way to use connection relationships, introducing both discriminant and redundant information. Thus it presents a trade-off between the advantages and disadvantages. We conducted an ablation study to examine its effects and report the results in Table 9. The additional structure features have positive effects on five datasets while others are negative. It doesn't significantly impact performance except for Roman-Empire. Moreover, CMGNN can still achieve competitive results without using additional structural features.

Table 9: Ablation study results (%) on the effects of additional structural features, where True denotes CMGNN with additional structural features and False denotes CMGNN with only node features.

| Structural Features | Roman-Empire | Amazon-Ratings | Chameleon-F | Squirrel-F | Actor | Flickr | BlogCatalog | Wikics | Pubmed | Photo |
|---|---|---|---|---|---|---|---|---|---|---|
| True | 68.43 ± 2.23 | **52.13 ± 0.55** | **45.70 ± 4.92** | **41.89 ± 2.34** | 35.72 ± 0.75 | **92.66 ± 0.46** | 96.47 ± 0.58 | **84.50 ± 0.73** | 88.90 ± 0.45 | 95.08 ± 0.43 |
| False | **84.35 ± 1.27** | 51.41 ± 0.57 | 44.85 ± 5.64 | 40.49 ± 1.55 | **36.82 ± 0.78** | 92.05 ± 0.75 | **97.00 ± 0.52** | 83.88 ± 0.75 | **89.99 ± 0.32** | **95.48 ± 0.29** |

### H.2.4 VISUALIZATION OF COMPATIBILITY MATRIX ESTIMATION

We visualize the observed and estimated CMs by CMGNN in Figure 6 with heat maps. Obviously, CMGNN estimates CMs that are very close to those existing in graphs. This shows that even with incomplete node labels, CMGNN can estimate high-quality CMs which provides valuable neighborhood information to nodes. Meanwhile, it can adapt to graphs with various levels of heterophily.

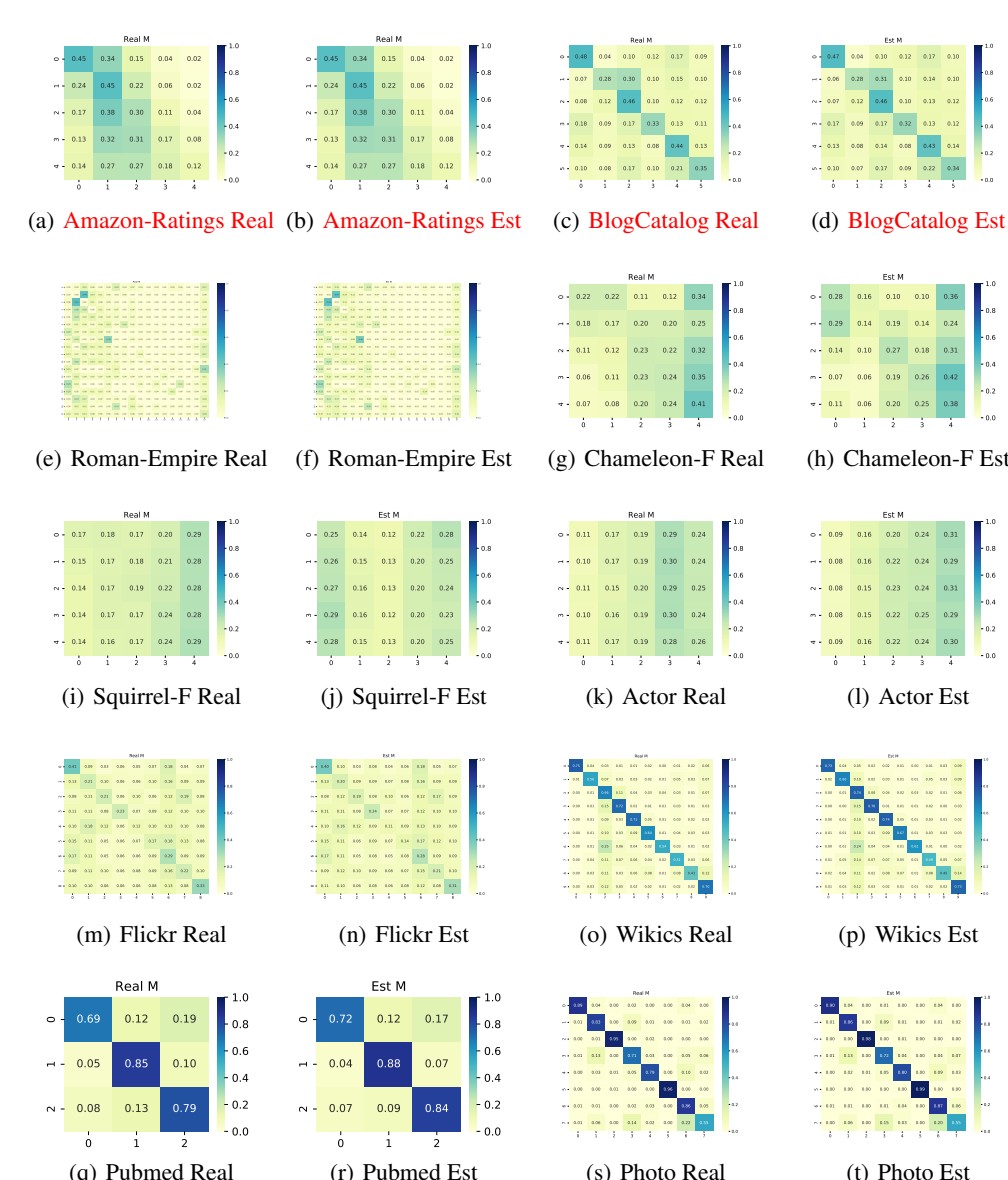

(a) Amazon-Ratings Real (b) Amazon-Ratings Est (c) BlogCatalog Real (d) BlogCatalog Est

(e) Roman-Empire Real (f) Roman-Empire Est (g) Chameleon-F Real (h) Chameleon-F Est

(i) Squirrel-F Real (j) Squirrel-F Est (k) Actor Real (l) Actor Est

(m) Flickr Real (n) Flickr Est (o) Wikics Real (p) Wikics Est

(q) Pubmed Real (r) Pubmed Est (s) Photo Real (t) Photo Est

Figure 6: The visualization of real and estimated CMs on other datasets.

### H.2.5 ADDITIONAL PERFORMANCE ON NODES WITH VARIOUS LEVELS OF DEGREES.

We show the additional performance on nodes with various degrees in Table 10. The results show that CMGNN can achieve relatively good performance on low-degree nodes, especially on heterophilous graphs. For the opposite results on homophilous graphs, we guess it may be due to the low-degree nodes in homophilous graphs having a more discriminative semantic neighborhood, such as a one-hot form. On the contrary, there are relatively more high-degree nodes with confused neighborhoods due to the randomness, which leads to the shown results on homophilous graphs.

Table 10: Node classification accuracy comparison (%) among nodes with different degrees.

| Dataset | Roman-Empire | | | | | Chameleon-F | | | | | Actor | | | | |
|---|---|---|---|---|---|---|---|---|---|---|---|---|---|---|---|
| Deg. Prop.(%) | 0~20 | 20~40 | 40~60 | 60~80 | 80~100 | 0~20 | 20~40 | 40~60 | 60~80 | 80~100 | 0~20 | 20~40 | 40~60 | 60~80 | 80~100 |
| **CMGNN** | **88.60** | **87.00** | **85.59** | **86.25** | **74.33** | 40.73 | **45.28** | **56.02** | **46.64** | 39.93 | 35.56 | 37.14 | 38.40 | 36.03 | 36.84 |
| ACM-GCN | 79.00 | 77.87 | 73.52 | 72.09 | 53.77 | 39.51 | 41.21 | 52.25 | 45.80 | **47.09** | 34.48 | 36.58 | 36.27 | 34.63 | **37.46** |
| OrderedGNN | **88.60** | **87.00** | 85.56 | 84.68 | 69.69 | **43.21** | 44.51 | 49.16 | 38.27 | 32.23 | 35.94 | **38.06** | 37.87 | 35.77 | 37.15 |
| GCNII | 86.79 | 85.14 | 85.20 | 84.75 | 71.09 | 34.84 | 42.56 | 47.50 | 40.45 | 41.84 | **36.89** | 37.20 | **38.53** | **38.02** | 36.99 |

| Dataset | Squirrel | | | | | Pubmed | | | | | Photo | | | | |
|---|---|---|---|---|---|---|---|---|---|---|---|---|---|---|---|
| Deg. Prop.(%) | 0~20 | 20~40 | 40~60 | 60~80 | 80~100 | 0~20 | 20~40 | 40~60 | 60~80 | 80~100 | 0~20 | 20~40 | 40~60 | 60~80 | 80~100 |
| **CMGNN** | **45.37** | **47.10** | **45.25** | **34.86** | 37.10 | 89.32 | 89.33 | 89.31 | **92.62** | **89.39** | 88.88 | **95.76** | 96.96 | **98.27** | 97.55 |
| ACM-GCN | 41.12 | 44.30 | 44.22 | 32.97 | **42.10** | 89.60 | **89.54** | 89.58 | 92.02 | 89.23 | 89.88 | 95.20 | 96.95 | 98.00 | 97.56 |
| OrderedGNN | 43.78 | 45.53 | 43.09 | 27.90 | 28.48 | 89.67 | 89.37 | 89.45 | 92.54 | 89.02 | **90.13** | 95.77 | 97.14 | 98.24 | **97.58** |
| GCNII | 43.08 | 45.55 | 43.65 | 33.07 | 38.05 | **89.77** | 89.50 | 89.24 | 92.45 | 88.86 | 88.89 | 95.36 | 97.12 | 97.83 | 96.64 |

## H.3 EFFICIENCY STUDY

**Complexity Analysis.** The number of learnable parameters in layer $l$ of CMGNN is $3d_r(d_r + 1) + 9$, compared to $d_r d_r$ in GCN and $3d_r(d_r + 1) + 9$ in ACM-GCN, where $d_r$ is the dimension of representations. The time complexity of layer $l$ is composed of three parts:

(i) AGGREGATE function: $O(Nd_r^2)$, $O(Nd_r^2 + Md_r)$ and $O(Nd_r^2 + NKd_r)$ for identity neighborhood, raw neighborhood and the supplementary neighborhood respectively, where $N$ and $M = |\mathcal{E}|$ denotes the number of nodes and edges;

(ii) COMBINE function: $O(3N(3d_r + 1) + 12N)$ for calculating adaptive weights and $O(3N)$ for combination;

(iii) FUSE function: $O(1)$ for concatenations.

Thus, the overall time complexity of $L$-layer CMGNN is $O(L(Nd_r(3d_r + K + 9) + Md_r + 18N) + 1)$, or $O(LNd_r^2 + LMd_r)$ for brevity.

**Experimental Running Time.** we report the actual average running time (ms per epoch) of baseline methods and CMGNN in Table 11 for comparison. The results demonstrate that CMGNN can balance both performance effectiveness and running efficiency.

Table 11: Effiency study results of average model running time (ms/epoch). OOM denotes out-of-memory error during the model training.

| Method | Roman-Empire | Amazon-Ratings | Chameleon-F | Squirrel-F | Actor | Flickr | BlogCatalog | Wikics | Pubmed | Photo |
|---|---|---|---|---|---|---|---|---|---|---|
| MLP | 7.8 | 7.0 | 6.1 | 6.5 | 6.3 | 9.1 | 6.7 | 6.4 | 6.1 | 5.8 |
| GCN | 33.8 | 33.4 | 7.9 | 20.6 | 34.4 | 37.2 | 30.4 | 25.5 | 35.6 | 28.1 |
| GAT | 15.9 | 67.3 | 10.3 | 14.0 | 30.8 | 66.2 | 17.6 | 26.8 | 33.4 | 36.0 |
| APPNP | 14.6 | 15.9 | 13.9 | 21.3 | 14.6 | 20.2 | 23.2 | 16.2 | 21.2 | 15.5 |
| GCNII | 29.4 | 28.4 | 37.3 | 19.6 | 37.7 | 84.2 | 97.6 | 20.7 | 258.0 | 46.9 |
| CPGNN | 12.7 | 20.3 | 12.2 | 13.4 | 13.6 | 18.9 | 16.7 | 15.5 | 14.0 | 11.7 |
| H2GCN | 20.0 | 31.2 | 17.2 | 32.4 | 55.6 | 415.7 | 165.5 | 332.8 | 39.0 | 87.6 |
| MixHop | 434.6 | 486.3 | 21.9 | 31.0 | 30.6 | 90.4 | 81.6 | 277.4 | 89.5 | 172.2 |
| GBK-GNN | 119.8 | 191.8 | 31.0 | 238.1 | 157.9 | OOM | OOM | 198.6 | 137.0 | 193.3 |
| GGCN | OOM | OOM | 55.7 | 42.1 | 199.8 | 111.2 | 108.7 | 226.6 | 2290.8 | 105.2 |
| GloGNN | 25.4 | 19.3 | 121.8 | 23.3 | 1292 | 562.9 | 30.9 | 1658.1 | 43.2 | 677.4 |
| HOGGCN | OOM | OOM | 25.2 | 54.3 | 1002.9 | 707.3 | 367.4 | 1406 | OOM | 655.3 |
| GPR-GNN | 15.9 | 12.5 | 22.3 | 23.2 | 16.7 | 15.9 | 14.7 | 49.8 | 13.2 | 13.1 |
| ACM-GCN | 56.7 | 56.7 | 26.1 | 29.7 | 22.5 | 60.7 | 31.7 | 42.4 | 37.1 | 40.1 |
| OrderedGNN | 86.0 | 110.8 | 49.5 | 60.1 | 67.8 | 107.0 | 88.3 | 116.9 | 88.1 | 78.2 |
| **CMGNN** | 51.5 | 93.5 | 62.5 | 64.7 | 19.0 | 52.5 | 69.8 | 44.0 | 102.9 | 20.4 |

**Trade-off Analysis between Effectiveness and Efficiency**. We have also visualized the trade-off between performance accuracy and empirical runtime compared to baseline methods in Figure H.3. From the results, we can see that CMGNN achieves the best performance with relatively low time consumption. Compared with OrderedGNN and GCNII, which have the second- and third-best average ranks, CMGNN offers both better accuracy and lower time consumption.

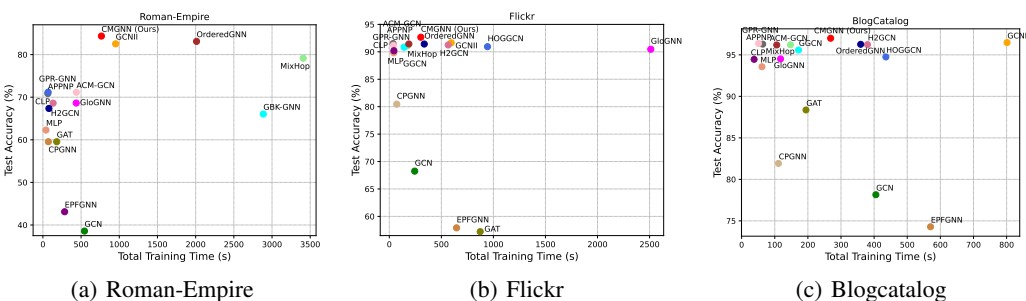

(a) Roman-Empire          (b) Flickr          (c) Blogcatalog

Figure 7: Visualizations of the trade-off between performance accuracy and training time compared with baseline methods on three representative datasets.

