# OpenReview forum: "Exploring and Unleashing the Power of Message Passing on Heterophilous Graphs"
_ICLR.cc/2025/Conference — Submitted to ICLR 2025_

### Official Review · Reviewer_cQFa · 2024-10-30

**Soundness:** 2
**Presentation:** 2
**Contribution:** 2
**Rating:** 3
**Confidence:** 3

**Summary:**

The paper studies message passing on heterophilic graphs and claims that the compatibility matrix is the key to the message-passing mechanism. Based on the compatibility matrix, the authors propose a new framework (i.e., CMGNN). The experiments show that CMGNN is superior.

**Strengths:**

- The experiments in this paper show the superiority of the proposed method.

**Weaknesses:**

- The paper lacks explanations in many key places, such as why A^{sup} and B^{sup} can supplement effective neighbor information. And some operations are wrong, such as the A^{sup} and B^{sup} \in R^{N*K}, while the Z is  R^{N*dr}, which can not conduct the inner-product operation in Eq. (9).
- Why A^{sup} is set to all 1? The authors need to explain.
- This paper is hard to read. Symbols are confusing, and symbols that appear for the first time are not explained. For example, what is H in Eq.(5)? what is \wave_{C} in Eq.(6)?

Overall, this paper is not solid, the motivation lacks verification, and the method is not clearly described. I tend to reject this paper.

**Questions:**

See weaknesses.

---

> ### Author Response · Authors · 2024-11-27
>
> We would like to thank you for your deeply thorough review. We have carefully considered your comments and suggestions, and the following are our detailed responses.
>
>
> > Weakness #1: Explanations about Eq.9.
>
> Answer #1: Thank you for pointing out this error. We initially described the message passing of the supplementary neighborhood **in text**, but in the revised version, we have added a separate **formula** to describe it more accurately. Specifically, the aggregated message $\mathbf{Z}^{l-1}r$ in the supplementary neighborhood is replaced by $\mathbf{Z}^{l-1}{ptt}$, which represents the virtual prototype nodes. These virtual nodes are obtained through the same message-passing mechanism as the real nodes. Therefore, $\mathbf{A}^{sup}, \mathbf{B}^{sup} \in \mathbb{R}^{N \times K}$ and $\mathbf{Z}^{l-1}_{ptt} \in \mathbb{R}^{K \times d_r}$ satisfy the inner product requirements.
>
> > Weakness #1 & Weakness #2: Explanations about $\mathbf{A}^{sup}$ and $\mathbf{B}^{sup}$.
>
> Answer #2: The supplementary neighborhood indicator $\mathbf{A}^{sup}$ contains $K$ additional prototype nodes. Given the sparsity of graphs, some nodes may have **low degrees**, and the all-one neighborhood indicator ensures that each node has access to messages from all classes. Meanwhile, the supplementary aggregation guidance, $\mathbf{B}^{sup} = \hat{\mathbf{C}} \hat{\mathbf{M}}$, specifies the desired semantic neighborhood of the nodes, i.e., the desired proportion of neighbors from each class, according to the probability that nodes belong to each class. As a result, $\mathbf{A}^{sup} \odot \mathbf{B}^{sup}$ provides each node with the desired neighborhood messages based on the predicted labels.
>
> > Weakness #3: Explanations about $\text{H}$ in Eq.5 and $\tilde{\mathbf{C}}$ in Eq.6.
>
> Answer #3: $H(p) = -\sum_i p_i \log(p_i)$ represents the information entropy. $\tilde{\mathbf{C}}$ refers to the predicted soft labels generated by the model, which is described in Eq.12 (or Eq.14 in the revised version).
>
>
>
> We are expecting these could be helpful in answering your questions.
> If you have any further questions, please feel free to contact us.

---

### Official Review · Reviewer_k6ce · 2024-10-31

**Soundness:** 2
**Presentation:** 2
**Contribution:** 2
**Rating:** 5
**Confidence:** 5

**Summary:**

The paper titled addresses the challenge of using GNNs on heterophilous graphs, where connected nodes often belong to different classes, defying traditional GNNs' homophily-based assumptions. It proposes the HTMP mechanism, a unified framework that improves message passing for such graphs by enhancing the compatibility matrix between classes. Additionally, the paper introduces CMGNN, a new GNN model under HTMP that explicitly leverages and refines the compatibility matrix to overcome issues related to noisy and incomplete semantic neighborhoods. Extensive benchmarking across various datasets and comparisons with different baselines validate CMGNN's effectiveness and the benefits of the HTMP approach in heterophilous settings.

**Strengths:**

1. **Unified Framework**: HTMP provides a cohesive structure for message passing in heterophilous GNNs, enabling flexibility and adaptability across different graph structures.

2. **Compatibility Matrix Enhancement**: CMGNN explicitly optimizes the compatibility matrix, effectively improving message relevance and class separability on heterophilous graphs.

3. **Comprehensive Benchmarking**: Extensive evaluations on 10 diverse datasets against 17 baselines validate CMGNN’s superior performance, demonstrating robustness in real-world heterophilous scenarios.

**Weaknesses:**

1. My primary concern with this paper is the Compatibility Matrix (CM). While the CM captures the likelihood of connections between classes in a graph, aiding message-passing in heterophilous GNNs, it closely resembles the attention mechanism by determining importance between different classes, rather than nodes. Additionally, the concept is not novel, as it has been employed in prior work, such as CPGNN [1]. Could the authors clarify the key differences introduced in this paper?

2. Although this paper focuses on message-passing GNNs, recent advances in spectral GNNs have shown competitive performance on heterophilous graphs. Were comparisons with spectral baselines like BernNet [2] and ChebNetII [3] considered? If spectral GNNs demonstrate superior performance, what is the rationale for focusing on message-passing GNNs?

3. In Table 3, “CMGNN” achieves identical performance to “W/O DL” on three datasets, including identical standard deviations, which seems unlikely given the exclusion of the “DL” component. Could the authors provide insights into this outcome?

4. Figure 8 is not in PDF format, making it appear blurred when zoomed in. A PDF version would enhance clarity.


[1] Zhu et al. Graph Neural Networks with Heterophily
[2] BernNet: Learning Arbitrary Graph Spectral Filters via Bernstein Approximation
[3] Convolutional Neural Networks on Graphs with Chebyshev Approximation, Revisited

**Questions:**

See Weaknesses

---

> ### Author Response · Authors · 2024-11-27
>
> We would like to thank you for your deeply thorough review. We have carefully considered your comments and suggestions, and the following are our detailed responses.
>
>
> > Weakness 1: Concern about the compatibility matrix.
>
> Answer #1: Thank you for your valuable comments. Indeed, the compatibility matrix models the connection preferences among classes rather than nodes, which may create a **gap** between the CM and the neighborhood of a single node due to **sparsity and noise** in real-world graphs, as discussed at the end of Section 4. Some existing methods use the compatibility matrix to redefine pairwise relations (i.e., edge weights) for existing edges, such as label propagation in CLP [1], log-likelihood estimation in EPFGNN [2], and prior belief propagation in CPGNN [3]. However, these methods still suffer from sparsity and noise. In contrast, CMGNN leverages the compatibility matrix and **virtual neighbors** to construct **supplementary messages** while **preserving the original neighborhood distribution in the CM**, resulting in better performance. A more detailed qualitative and quantitative comparison and analysis between CMGNN and existing CM-based methods are available in Section 6.3 and Appendix H.2.1.
>
> References:
>
> [1] Simplifying Node Classification on Heterophilous Graphs with Compatible Label Propagation.
>
> [2] Explicit Pairwise Factorized Graph Neural Network for Semi-supervised Node Classification.
>
> [3] Graph Neural Networks with Heterophily.
>
> > Weakness #2: Comparisons with spectral GNNs.
>
> Answer #2: Thank you for your suggestion. On the one hand, the message-passing mechanism plays a significant role in existing heterophilous GNNs and is relatively straightforward to understand. Therefore, we proposed the HTMP mechanism to provide a unified understanding of several representative methods from a message-passing perspective. On the other hand, we have also reviewed some relatively simple spectral GNNs and understood them from a message-passing perspective, such as FAGCN and ACMGCN. For more complex methods like BernNet and ChebNetII, we will consider conducting experiments to compare their performance and evaluate the feasibility of understanding them from the message-passing perspective.
>
> > Weakness #3: Explanations about the results of the ablation study.
>
> Answer #3: In the implementation of CMGNN, the weight parameter of the DL loss, denoted as $\lambda$, has a search space that includes 0, as shown in Table 7. The best choice of $\lambda$ for these three datasets is **0**, which results in identical performance for both the "CMGNN" and "W/O DL" settings. This may be due to the discriminability of the desired neighborhood messages reaching a bottleneck, where further improvement by DL is no longer possible.
>
> > Weakness #4: Format of Figure 8.
>
> Answer #4: Thank you for this suggestion! We have updated it to PDF format in the revised version.
>
>
>
> We are expecting these could be helpful in answering your questions.
> If you have any further questions, please feel free to contact us.

---

### Official Review · Reviewer_vfyJ · 2024-11-02

**Soundness:** 2
**Presentation:** 2
**Contribution:** 2
**Rating:** 3
**Confidence:** 4

**Summary:**

This paper examines the effectiveness of message-passing mechanisms in heterophilous graph neural networks (HTGNNs), where connected nodes often display contrasting behaviors. This paper propose a unified heterophilous message-passing (HTMP) mechanism and introduce CMGNN, which enhances the compatibility matrix to address incomplete and noisy semantic neighborhoods.

**Strengths:**

1. Building a unified codebase for heterophily benchmarks/baselines would greatly benefit community research.

**Weaknesses:**

1. **Motivation**: The HTMP framework proposed in this paper does not appear to specifically address the heterophily problem, as the mentioned components like FUSE and COMBINE are generally applicable (for instance, they are used in many works focusing on over-smoothing or over-squashing). Fundamentally, they are not exclusive to the heterophily issue and resemble more a general extension of the message-passing mechanism.

2. **Experiments**: The experiments in this paper are conducted only on datasets with a relatively small node scale (with a maximum of just over 20,000 nodes), which does not demonstrate the scalability of the proposed method.

3. **Methodology**

   - The paper's proposed CMGNN does not clarify the specific reasons for selecting components in AGG, COMBINE, and FUSE, namely how your chosen designs can help the proposed HTMP better estimate the compatibility matrix.

   - Moreover, the design of these components appears overly manual (e.g., why combine from only the three proposed types of neighbors in COMBINE? Why use concatenation in FUSE but weighted summation in COMBINE?), seeming like a patchwork of existing methods.

   - Finally, regarding using the rows of the adjacency matrix as additional node features, I believe this could lead to the model being unscalable (as the number of model parameters increases with the number of nodes) and seems to disrupt the permutation invariance of GNN models.

**Questions:**

1. Authors mentioned that during training, for efficiency, they only update the compatibility matrix when evaluation performance improves. How much difference in computation time is there between epochs when the CM is updated and when it is not?

---

> ### Author Response · Authors · 2024-11-27
>
> We would like to thank you for your deeply thorough review. We have carefully considered your comments and suggestions, and the following are our detailed responses.
>
>
> > Weakness #1: The association between the HTMP mechanism and the heterophily issue.
>
> Answer #1: Indeed, HTMP is an extension of the vanilla message-passing mechanism. Some components, like the FUSE and COMBINE functions, may be applicable to other topics related to message passing. However, **certain specific designs are exclusive to heterophily**, such as ego-neighbor separation and negative adaptive weight. On the other hand, some works have found that the **over-smoothing issue mentioned shares the same root causes as heterophily** and can be addressed with similar approaches. Thus, it is reasonable that similar designs exist in both cases.
>
> > Weakness #2: The scalability of CMGNN.
>
> Answer #2: Thank you for pointing this out. To further evaluate the scalability of CMGNN, we conducted additional experiments on three large-scale datasets, comparing its performance with representative baselines. The details of the datasets and the results are listed as follows (or in Table 8 of the revised version). As a result, CMGNN achieves superior performance while maintaining good efficiency, especially on snap-patents, with a **22% improvement**, which demonstrates its great scalability.
>
> |**Dataset**| **snap-patents**| **snap-patents**|**pokec**| **pokec**| **genius**|**genius**|
> |-|-|-|-|-|-|-|
> | **Homo.**|0.073||0.445||0.618||
> | **Nodes**|2,923,922||1,632,803||421,961||
> | **Edges**|13,975,788||30,622,564||984,979 ||
> | **Classes**|5||2||2||
> | **Method**|**acc (%)**|**cost (min)**|**acc (%)**|**cost (min)**|**acc (%)**|**cost (min)**|
> | **MLP**|31.30 ± 0.07|37|62.29 ± 0.09|75|82.54 ± 0.14|2|
> | **GCN**|37.97 ± 0.04| 87| 70.12 ± 0.13| 96| 84.25 ± 0.13| 12|
> | **GAT** | 38.42 ± 0.24 | 80 | 73.76 ± 0.38 | 287| 81.89 ± 0.39| 16|
> | **OrderedGNN** | 38.43 ± 0.22| 305| 75.17 ± 0.18| 273| _85.15 ± 0.65_| 63 |
> | **GCNII** | _40.90 ± 0.18_ | 562| _78.18 ± 0.49_ | 522 | 84.99 ± 0.48  | 188  |
> | **CMGNN** | **62.86 ± 0.38** | 148 | **81.74 ± 0.50**| 393| **85.44 ± 0.20**| 27 |
>
>
>
>
> > Weakness #3: The underlying reasons for component design in CMGNN.
>
> Answer #3: CMGNN aggregates messages from three neighborhoods for each node: the ego, raw, and supplementary neighborhoods. The first two are the most commonly used and contain information about the central node itself and its one-hop neighbors, respectively, while the latter introduces **a new way to utilize the compatibility matrix**. Given the **diverse situations** of different nodes, we use adaptive weighted addition to combine the messages from the three neighborhoods. Additionally, the messages from multiple layers are concatenated to preserve information with **varying levels of locality** in the graph.
>
> > Weakness #4: Problems caused by additional structural features.
>
> Answer #4: Utilizing additional structural features is a common approach in heterophilous GNNs, as it offers another way to leverage connection relationships, introducing both discriminative and redundant information. This creates a trade-off between the advantages and disadvantages. In CMGNN, the use of additional structural features is **optional**, meaning they can be excluded in large-scale graphs to ensure scalability. Additionally, we have conducted an ablation study to examine the effects of these features, with results reported in Appendix H.2.2 (or H.2.3 in the revised version). CMGNN can still achieve competitive results even without using additional structural features. The results in Answer #2, which do not use these features, also support this conclusion.
>
> > Question #1: The difference in computation time between updating CM per epoch or not.
>
> Answer #5: We conducted a comparison of computation times and show the results below, where "CMGNN*" refers to the identical model, except that the CM is updated per epoch. According to the results, the CM update strategy reduces computation time by up to **25%** (on Pubmed) while maintaining similar performance.
>
> | **Method**   | **Roman-Empire**   | **Amazon-Rating**   | **Chameleon-F**  | **Squirrel-F**   | **Actor**   | **Flickr**  | **BlogCatalog**  | **Wikics**   | **Pubmed**   | **Photo**  |
> |-----------|---------|--------|----------|---------|--------|---------|---------|-----------|-----------|---------|
> |**CMGNN**   | 51.5   | 93.5   | 62.5   | 64.7   | 19.0   | 52.5   | 69.8   | 44.0   | 102.9   | 20.4 |
> |**CMGNN\***  | 56.7   | 104.0   | 80.4  | 85.7   | 22.9   | 55.4   | 81.4   | 43.8   | 136.9   | 23.4 |
>
>
> We are expecting these could be helpful in answering your questions.
> If you have any further questions, please feel free to contact us.

---

### Official Review · Reviewer_qZEx · 2024-11-06

**Soundness:** 2
**Presentation:** 2
**Contribution:** 2
**Rating:** 3
**Confidence:** 4

**Summary:**

This paper proposes a compatibility matrix-enhanced method to improve the expressiveness of graph neural networks on heterophilous graph data, and verifies its validity on 10 benchmark datsets.

**Strengths:**

Extensive experiments have been conducted, including empirical studies on compatibility matrices for various models.

The experiments may be convincing based on the demonstration of the details.

**Weaknesses:**

1. The claim that success of HTGNN is due to "enhancing the compability matrix" is problematical. The compability matrix depicts the relations between classes whose instances are linked together, so naturally better representation (thus better prediction) definitely results in more desirable compability matrix. It seems that compability matrix should be the effect, not the cause.  It would be good to provide more evidence or analysis demonstrating that enhancing the compatibility matrix directly leads to improved representations, rather than being a consequence of them.
2. The theorems and lemmas (e.g., theorem 1) are overlay simple. Most of the results are obvious, without meaningful insights.
3. The motivation of "supplementary neighborhood construction" is not explained in Methodology.
4. No time and space complexity analysis of the proposed method.
5. The claim on "New datasets" introduced by this paper is inappropriate. It should precisely describe what is new about the dataset usage compared to previous work (unified data splitting cannot be the "new"), such as modifications to existing datasets, new combinations, or different preprocessing steps that the authors consider novel. that performs well on both homophilous and heterophilious graphs,
6. why some popular SOTAs like graphSAGE and recently developed methods such as Ref. [1-2]  are not included in comparison? The former performs well on both homophilous and heterophilious graphs, the latter two are also oriented towards heterophily issue.
7. The writing needs improvement. There are some  vague sentences and  non-standard writings,  eg. 232-234, 'it's', 'nodes' neighborhood'.

Ref. [1] Wang et al. "understanding heterophily of graph neural networks", ICML 2024
 [2] LG-GNN, IJCAI 2024.

**Questions:**

1. Line 59-60, why increasing distinctiveness between rows of compability matrix will enhance node representation? It should be justified. Throughout the paper, I can only see the emphasis on the distinction between diagonal and non-diagonals of compability matrix.
2. Why design weights in the form of Eq. 7? no any intuitions behind.
3. Despite of the supplementary neighborhood construction (which can be viewed as additional label propagation) and the compability matrix optimization component, the performance improvement is still marginal, compared to the baselines in the paper.
4. what does it mean by "k neighborhoods"? how to count the number of neighborhoods for a node in question?

---

> ### Author Response · Authors · 2024-11-27
>
> We would like to thank you for your deeply thorough review. We have carefully considered your comments and suggestions, and the following are our detailed responses.
> > W1 & Q1: The causal relationship between compatibility matrix and representations.
>
> A1: The compatibility matrix (CM) is used to represent the connection preferences inherent in the graph, which are computed from **node labels and edges** rather than from representations or model predictions. Therefore, better representations are not the cause of a more desirable CM. Theorem 1 shows a positive correlation between the discriminability of CM and the discriminability of representations in the message-passing mechanism. Thus, existing heterophilous methods can enhance the CM in various ways (e.g., by modifying the aggregated edge weights) to learn better representations.
> > W2: The theorems and lemmas are overly simple.
>
> A2: Indeed, the theorems and lemmas are simple, but they lead to interesting conclusions about the CM. Existing works treat CM as a fixed property of graph data, but we argue that it can be **enhanced** for heterophilous message passing to learn better representations.
> > W3: The motivation for "supplementary neighborhood construction" is not explained.
>
> A3: Considering the sparsity of graphs, some nodes may have low degrees, limiting the potential of CM. Thus, we construct supplementary neighborhoods to guarantee **accessibility to messages from each class** for all nodes. We have added more detailed descriptions of this motivation in the revised version.
> > W4: No time and space complexity analysis of the proposed method.
>
> A4: Due to page limitations, the efficiency study, including computational and memory complexity analysis as well as empirical runtime comparisons, is provided in Appendix H.2.5 (or H.3 in the revised version).
> > W5: The claim about "New datasets" is inappropriate.
>
> A5: We have revised the wording to "Newly organized datasets" in the revised version. Specifically, to address the issues of method comparison caused by drawbacks like data leakage in existing datasets, we have recollected and filtered suitable graph datasets. This collection spans various levels of homophily, providing a robust foundation for performance evaluation. However, the benchmark datasets are not the main contribution of the paper, and we have not modified the details of the datasets. Thus, we have corrected the description accordingly.
> > W6: Supplement of baseline methods.
>
> A6: Thank you for your suggestion. We will consider adding them to the baseline methods for comparison in further revised versions, due to the limited time and computational resources.
> > W7: The writing needs improvement.
>
> A7: Thank you for the feedback. We have revised some of the expressions and will continue to improve the writing in future revisions.
> > Q2: Why design weights in the form of Eq.7?
>
> A8: Eq.7 defines a weighting function that accounts for the effects of node degrees. The core idea is that nodes with lower degrees should correspond to lower weights in the CM estimation, as high-degree nodes usually have more representative neighborhoods, while low-degree nodes often have incomplete ones. Meanwhile, the relationship between weights and node degrees is not linear. For low-degree nodes, increases in degree should yield more significant benefits compared to high-degree nodes. Beyond a certain threshold, increases in degree offer diminishing returns. We empirically chose K and 3K as fixed thresholds for the weighting function to simplify the design without extensive trials. This approach is straightforward and can be substituted with any other forms that meet the same criteria. While further design is possible, it is not a priority compared to other modules.
> > Q3: The performance improvement is marginal compared to the baselines.
>
> A9: The main contribution of this paper is to provide a unified theoretical framework for existing heterophilous GNNs and to re-understand the underlying mechanism. Guided by the HTMP mechanism, CMGNN has a simple architecture and outperforms existing complex methods. Furthermore, we conducted additional experiments on large-scale graphs with results shown in Table 8 of the revised version. The performance and efficiency comparison demonstrates the effectiveness of CMGNN, particularly on snap-patents (with a **22% improvement**).
> > Q4: What does "k neighborhoods" mean? How do you count the number of neighborhoods for a node in question?
>
> A10: We apologize for the confusion, but we are unsure what "k neighborhoods" refers to, as this term does not appear in the paper. If you are referring to the "R neighborhoods" mentioned in Section 3, this refers to the multiple neighborhoods used by heterophilous GNNs for message passing, such as raw, high-order, and similarity-based neighborhoods.
>
> We are expecting these could be helpful in answering your questions.
> If you have any further questions, please feel free to contact us.

---

> > ### Comment · Reviewer_qZEx · 2024-11-29
> >
> > Thank you for your response. Regarding A1,  CM is directedly obtained from node labels , as the authors demonstrated, but since most of node labels are invisible (except a few in common semi-supervised setting) in training, CM is not known a prior, How CM acts on node representation learning (i.e., managing the  learning process) needs to be clearly articulated. If I understood right, current strategy the paper proposed is to alternatively update representation and CM, but it requires representations being available at the first place, which conflicts with the claim that CM is the cause.
> >
> > Regarding A10, I suggest that using k-order to remove the ambiguity of "multiple neighborhoods".

---

> > > ### Author Response · Authors · 2024-11-29
> > >
> > > Thank you for your response!
> > >
> > > In CMGNN, the compatibility matrix is estimated using soft pseudo labels, which are composed of training labels and model predictions. Initially, the model lacks predictions, so the model prediction $\tilde{\mathbf{C}}$ is manually initialized with values of $\frac{1}{k}$, where $k$ is the number of classes. The compatibility matrix is then estimated for the first time without utilizing representations. Subsequently, it can be further used to learn representations.
> > >
> > > Regarding the ambiguity of "multiple neighborhoods," we appreciate your kind suggestion. However, the K-order neighborhood is only one example of multiple neighborhoods and does not fully capture this concept. It also includes other types of neighborhoods, such as those based on similarity or custom-defined relationships.
> > >
> > > If you have any additional questions, please don’t hesitate to contact us.

---

> > > > ### Comment · Reviewer_qZEx · 2024-11-29
> > > >
> > > > Thank you for the clarification.

---

### Meta-Review · Area_Chair_CWxB · 2024-12-11

**Metareview:**

This paper unifies existing GNNs for heterophilous graphs by extending message-passing with a compatibility matrix and presents a new method to estimate this matrix. Although it achieves new SOTA on some small datasets, the following issues make this paper below the acceptance threshold. Firstly, the novelty is limited since the compatibility matrix, such as the stochastic block model, has been widely used. Secondly, it lacks rigorous theoretical analysis to support it. Most are represented as observations. Thirdly, the motivation of "supplementary neighborhood construction" is not clear. Finally, the employed datasets are small to justify the effectiveness. The datasets introduced in LINKX should be considered.

**Additional Comments On Reviewer Discussion:**

Although the authors provided some feedback, reviewers also think that most concerns are not alleviated. Thus, all of them keep the negative ratings and believe it below the acceptance threshold.

---

### Decision · Program_Chairs · 2025-01-22

Reject